# Semantic-aware Adversarial Fine-tuning for CLIP

**Jiacheng Zhang**  *jiacheng.zhang6@unimelb.edu.au*
*School of Computing and Information Systems*
*The University of Melbourne*

**Jinhao Li**  *jinhao.li2@unimelb.edu.au*
*School of Computing and Information Systems*
*The University of Melbourne*

**Hanxun Huang**  *curtis.huang1@unimelb.edu.au*
*School of Computing and Information Systems*
*The University of Melbourne*

**Sarah M. Erfani**  *sarah.erfani@unimelb.edu.au*
*School of Computing and Information Systems*
*The University of Melbourne*

**Benjamin I.P. Rubinstein**  *benjamin.rubinstein@unimelb.edu.au*
*School of Computing and Information Systems*
*The University of Melbourne*

**Feng Liu**  *feng.liu1@unimelb.edu.au*
*School of Computing and Information Systems*
*The University of Melbourne*

**Reviewed on OpenReview:** *[https://openreview.net/forum?id=SzZOBzueK0](https://openreview.net/forum?id=SzZOBzueK0)*

## Abstract

Recent studies have shown that CLIP model's adversarial robustness in zero-shot classification tasks can be enhanced by adversarially fine-tuning its image encoder with *adversarial examples* (AEs), which are generated by minimizing the *cosine similarity* between images and a hand-crafted template (e.g., "A photo of a {label}"). However, it has been shown that the cosine similarity between a single image and a single hand-crafted template is insufficient to measure the similarity for image-text pairs. Building on this, in this paper, we find that the AEs generated using cosine similarity may *fail to fool* CLIP when the similarity metric is replaced with semantically enriched alternatives, making the image encoder fine-tuned with these AEs less robust. To overcome this issue, we first propose a *semantic-ensemble attack* to generate semantic-aware AEs by minimizing the average similarity between the original image and an ensemble of *refined* textual descriptions. These descriptions are initially generated by a foundation model to capture core semantic features beyond hand-crafted templates and are then refined to reduce hallucinations. To this end, we propose **S**emantic-aware **A**dversarial **F**ine-**T**uning (SAFT), which fine-tunes CLIP's image encoder with semantic-aware AEs. Extensive experiments show that SAFT outperforms current methods, achieving substantial improvements in zero-shot adversarial robustness across 16 datasets. Our code is available at: [https://github.com/tmlr-group/SAFT](https://github.com/tmlr-group/SAFT).

## 1 Introduction

*Contrastive language-image pre-training* (CLIP) (Radford et al., 2021) is a widely adopted framework that learns to encode text and images into a unified feature space using large datasets. This approach

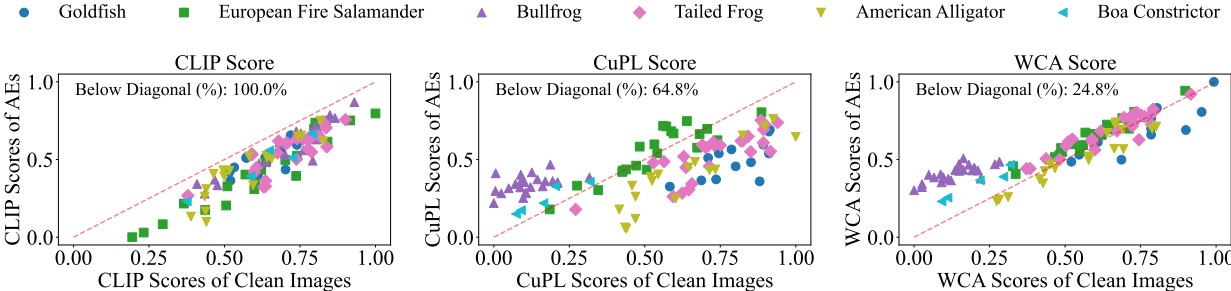

Figure 1: Comparison between CLIP (Radford et al., 2021), CuPL (Pratt et al., 2023) and WCA (Li et al., 2024) as similarity metrics for clean images and their corresponding AEs, generated by minimizing the CLIP score via *projected gradient descent* (PGD) (Madry et al., 2018), across six animal classes in ImageNet-1K (Deng et al., 2009). Points below the diagonal line indicate a success in attacking the similarity metric. The results show that although these AEs can reduce the CLIP score, they may *fail to fool* CLIP when more semantically enriched scores are used as alternatives. This observation motivates us to rethink how AEs should be constructed in the case of CLIP.

enables remarkable zero-shot generalization capabilities and has been applied across numerous downstream applications, including large *vision-language models* (VLMs) such as Flamingo (Alayrac et al., 2022) and LLaVA (Liu et al., 2023). However, despite the remarkable success of CLIP-based models across a wide range of downstream tasks, their vulnerability to *adversarial examples* (AEs) raises significant concerns about their safe deployment in real-world scenarios (Mao et al., 2023; Huang et al., 2025; Ma et al., 2025).

Fine-tuning the CLIP's image encoder with *adversarial training* (AT) has emerged as an effective approach to enhancing its adversarial robustness (Mao et al., 2023; Schlarmann et al., 2024; Wang et al., 2024a; Yu et al., 2024; Zhang et al., 2024). The core idea of AT is to improve model robustness by training on AEs that are generated dynamically during the training process (Madry et al., 2018). AEs are typically generated by maximizing the cross-entropy loss between the model's predicted class probability and the true label (Goodfellow et al., 2015; Madry et al., 2018). In the case of CLIP, this probability is derived from the cosine similarity between a single image and a hand-crafted text template (e.g., "A photo of a {label}"), commonly referred to as the CLIP score (Radford et al., 2021).

Recent studies (Menon & Vondrick, 2023; Pratt et al., 2023; Li et al., 2024) have shown that the CLIP score is often insufficient to fully capture image-text alignment. To address this, they propose semantically enriched similarity metrics as advanced alternatives to the CLIP score. For example, Pratt et al. (2023) propose *customized prompts via language models* (CuPL), which measures the similarity between images and LLM-generated class-specific descriptions (e.g., "A platypus looks like a beaver with a duck's bill" instead of "A photo of a platypus"). Li et al. (2024) further propose *weighted visual-text cross alignment* (WCA), which measures the similarity between localized image patches and CuPL-based descriptions. This naturally leads us to pose the following question:

*Are adversarial examples generated by minimizing the CLIP score truly effective in degrading image-text alignment when evaluated under more semantically enriched similarity metrics?*

In this paper, we find that these AEs may *fail to fool* CLIP when semantically enriched similarity metrics are used, as demonstrated by Figure 1. When CLIP score is used, 100% of points fall below the diagonal, indicating that these AEs successfully degrade the similarity. However, when the similarity metric is replaced by other alternatives such as WCA or CuPL, this degradation *diminishes significantly*: only 28.6% and 64.8% of AEs fall below the diagonal line under WCA and CuPL, respectively. More surprisingly, we observe that some AEs even achieve higher similarity scores than their clean counterparts (i.e., those above the diagonal), suggesting that CLIP becomes *more confident* in these inputs after the attack, in other words, these AEs somehow *assist* CLIP in making more confident predictions. This observation suggests that adversarial perturbations optimized by the CLIP score using a single hand-crafted template (e.g., "A photo of a label") often *fail* to generalize to alternative text descriptions with richer attributes or contexts, and thus make the

generated AEs less effective during adversarial fine-tuning. Eventually, it leads to a less robust image encoder, as the success of AT-based methods (e.g., adversarial fine-tuning) critically depends on the effectiveness and universality of AEs (Madry et al., 2018).

To overcome this issue, we propose **S**emantic-aware **A**dversarial **F**ine-**T**uning (SAFT), a new framework that aims to use more semantically enriched AEs to fine-tune the CLIP's image encoder. We first propose a *semantic-ensemble attack* method, which generates *semantic-aware AEs* by minimizing the average similarity between the original image and an ensemble of *refined* textual descriptions. These textual descriptions are initially generated by a foundation model (e.g., a LLM or a MLLM), aiming to encapsulate diverse attributes, contexts, and synonyms related to each class label.

However, foundation models are known to suffer from hallucinations (Maynez et al., 2020). For instance, given the class "dog", a foundation model might hallucinate that dog is a "winged mythical creature." Therefore, to make sure these descriptions are semantically relevant and factually correct, we further propose *hallucination-aware description generation* method, which retains only top-$K$ refined descriptions that are closely aligned with the class's core semantics based on the relevance score. During adversarial fine-tuning, we optimize the parameters of CLIP's image encoder to align AEs with a diverse set of textual descriptions selected through our hallucination-aware generation method. This encourages the encoder to map perturbed images into regions of the embedding space that are invariant to both visual perturbations and linguistic variations. We provide a visual illustration of SAFT in Figure 2 and an algorithmic description in Algorithm 1.

Through extensive experiments on 16 benchmark image datasets (including 1 in-domain dataset and 15 zero-shot datasets), we demonstrate the effectiveness of SAFT in Section 5. SAFT improves zero-shot robust accuracy over the current *state-of-the-art* (SOTA) methods by at least 3.85% on average, while also achieving the second-highest zero-shot clean accuracy across the 15 zero-shot datasets (see Table 1). In addition, we demonstrate that SAFT can scale to larger CLIP models (e.g., CLIP-L/14) in Table 4 and large datasets (e.g., ImageNet-1K) in Table 7, respectively. More importantly, we show that SAFT can generalize well to unseen text templates (see Table 10) and can be applied to other downstream tasks beyond classification such as the image-text retrieval task (see Table 5).

Our main contributions are: (1) we observe that the CLIP score, computed between a single image and a single handcrafted text template, is insufficient for accurately evaluating image-text similarity. Consequently, AEs generated by minimizing the CLIP score tend to be significantly less effective, leading to less robust image encoder. This is an aspect that has been largely overlooked in existing studies; (2) to address the above-mentioned limitation, we propose **S**emantic-aware **A**dversarial **F**ine-**T**uning (SAFT), a novel framework that generates AEs by minimizing the average similarity between an image and an ensemble of selected textual descriptions during the CLIP fine-tuning process. To mitigate potential hallucinations caused by LLMs, we further propose a semantic filtering method to help remove descriptions that deviate significantly from the intended semantic meaning; (3) we empirically show that, compared to the existing adversarial fine-tuning methods, SAFT achieves a notable improvement in zero-shot accuracy-robustness trade-off on 16 benchmark image datasets against multiple adversarial attacks and generalizes well to unseen text templates.

## 2 Related Work

**Adversarial Robustness.** The vulnerability of deep learning models to AEs has been a long-standing challenge in the community and has been extensively studied (Szegedy et al., 2014; Goodfellow et al., 2015; Carlini & Wagner, 2017; Ilyas et al., 2019; Croce et al., 2021; Huang et al., 2021; Zhang et al., 2025; Sun et al., 2025). AEs are typically generated by introducing imperceptible perturbations to clean images, with the objective of misleading a classifier into making incorrect predictions. *Adversarial training* (AT) (Goodfellow et al., 2015; Madry et al., 2018; Zhang et al., 2019; Wang et al., 2020; Wu et al., 2020; Liu et al., 2021; Zhang et al., 2021; 2024) is widely regarded as the most effective strategy for defending against AEs, particularly due to its resilience against adaptive attacks (Athalye et al., 2018). However, AT often degrades performance on clean examples, a phenomenon known as the accuracy-robustness trade-off (Tsipras et al., 2019). Addressing this trade-off and achieving a better balance remains a key focus in AT research. AT works by generating AEs and incorporating them into the model's training process, forcing the model to learn the underlying

distributions of AEs. However, most existing studies on AT focus on image classifiers trained with supervised learning and are known for being computationally expensive, making it challenging to scale for CLIP.

**Vision-language Models.** Recent advances in *vision-language models* (VLMs) have significantly improved multi-modal understanding by aligning visual and textual representations through large-scale pre-training. Radford et al. (2021) introduced CLIP, a pioneering model that employs contrastive learning on image-text pairs to enable zero-shot transfer across diverse tasks, demonstrating remarkable generalization. Pre-trained image encoders have been widely adopted in large VLMs (Alayrac et al., 2022; Awadalla et al., 2023; Wang et al., 2023; Bai et al., 2023; Liu et al., 2023; Zhu et al., 2024), which align the image encoder with LLMs in token embedding space via a bridging network or lightweight querying transformer (Li et al., 2023). The majority of large VLMs rely on CLIP as their pre-trained image encoder, primarily because it is trained with text supervision (Tong et al., 2024). While these large VLMs have achieved significant success across various tasks, their vulnerability to AEs (Mao et al., 2023; Schlarmann et al., 2024) raises critical concerns regarding their safe deployment. As a result, improving the robustness of CLIP has become an important challenge.

**Adversarial Finetuning for CLIP.** Fine-tuning the CLIP encoder with AT has emerged as a cost-effective approach to enhancing its adversarial robustness. TeCoA is the first method to improve the zero-shot robustness of VLMs, which fine-tunes CLIP encoders through AT (Mao et al., 2023). TeCoA uses a fixed zero-shot template with a class label to generate and train on AEs. FARE builds upon TeCoA by incorporating unsupervised objectives that maximize the distance in the embedding space to generate AEs (Schlarmann et al., 2024). PMG-AFT leverages auxiliary branches to minimize the embedding distance between outputs on AEs and clean examples in both the target and pre-trained models, mitigating adversarial overfitting (Wang et al., 2024a). TGA-ZSR introduces text-guided attention to further improve zero-shot robustness (Yu et al., 2024). Adversarial fine-tuning can also be extended to consider multimodal inputs (Zhou et al., 2024). Additionally, adversarial training can be performed from scratch during vision-language pretraining, though these approaches are computationally intensive (Gan et al., 2020; Wang et al., 2024b). In this work, our focus is adversarial fine-tuning for CLIP and we make the *first* attempt to show that enriched textual embeddings enable the generation of more generalizable AEs that are invariant to minor textual variations.

**Textual Prompting in Vision-language Models.** Although CLIP demonstrates strong zero-shot performance, its effectiveness on downstream tasks is highly dependent on prompt design, as highlighted by Radford et al. (2021) and Zhou et al. (2022). To mitigate this sensitivity, Menon & Vondrick (2023) and Pratt et al. (2023) propose to leverage the knowledge embedded in LLMs to automatically generate class-specific descriptions. Li et al. (2024) further propose to use localized visual prompting (e.g., random cropping) to obtain multiple patches representing local visual areas of the query image and then cross-align these local image patches with class-specific descriptions. Cai et al. (2025) propose to capture multiple common and unique features for each class by guiding foundation models to generate descriptive attributes and distinctive attributes. More recently, Sun et al. (2026) further improves existing methods (Pratt et al., 2023; Li et al., 2024) by proposing two complementary components to filter out redundant image crops using IoU overlap and remove repetitive or semantically similar textual descriptions via embedding-based cosine similarity. These semantically enriched textual descriptions have been shown effective for image-text alignment. However, to the best of our knowledge, whether such enriched textual descriptions improve the zero-shot accuracy–robustness trade-off remains underexplored.

In this work, motivated by our finding that AEs may *fail to fool* CLIP when alternative similarity metrics are used, we make the *first* attempt to show that semantically enriched textual descriptions can craft more generalizable AEs that are invariant to minor textual variations. We demonstrate that leveraging these AEs during adversarial fine-tuning further enhances the zero-shot accuracy-robustness trade-off.

## 3 Problem Setting and Preliminaries

### 3.1 Problem Setting

The *zero-shot adversarial robustness* problem, introduced by Mao et al. (2023), can be mathematically formulated as follows. Let $\mathfrak{T}$ denote a distribution over unseen classification tasks. Each task $\mathcal{T} \sim \mathfrak{T}$ defines a label space with $N$ classes and an associated data distribution $\mathcal{D}_{\mathcal{T}}$. An attacker, with full access to

task-specific ground-truth labels $y_{\mathcal{T}}$, crafts an AE within an $\ell_p$-bounded perturbation set $\Delta = \{\delta : \|\delta\|_p \leq \epsilon\}$ by maximizing $\mathcal{L}(f^\theta(x + \delta), y_{\mathcal{T}})$. In contrast, the defender lacks access to the task identity $\mathcal{T}$ or distribution $\mathcal{D}_{\mathcal{T}}$, and must train a model $f^\theta$ with parameters $\theta$ to minimize the expected worst-case risk over all tasks:

$$\min_\theta \; \mathbb{E}_{\mathcal{T} \sim \mathfrak{T}} \left[ \mathbb{E}_{(x, y_{\mathcal{T}}) \sim \mathcal{D}_{\mathcal{T}}} \max_{\delta \in \Delta} \mathcal{L}(f^\theta(x + \delta), y_{\mathcal{T}}) \right].$$

This formulation stands in contrast to standard adversarial robustness, which is typically evaluated on a single, known task $\mathcal{T}_0$.

## 3.2 Contrastive Language-Image Pre-training

*Contrastive language-image pre-training* (CLIP) (Radford et al., 2021) is a dual-encoder model consisting of an image encoder $f_{\text{img}}^\theta : \mathcal{I} \to \mathbb{R}^d$ and a text encoder $f_{\text{text}} : \mathcal{Z} \to \mathbb{R}^d$, where $\mathcal{I}$ and $\mathcal{Z}$ denote the input spaces of images and texts, and $d$ is the dimension of the shared embedding space. For zero-shot classification, CLIP uses a template-based prompting strategy. Given a label set $\mathcal{Y} = \{y_1, \ldots, y_N\}$, each label $y \in \mathcal{Y}$ is embedded into a textual prompt $p(y)$ (e.g., "A photo of a {label}"). At inference, given an image $x$, CLIP predicts its class by computing cosine similarities between the image embedding $f_{\text{img}}^\theta(x)$ and a set of text embeddings $\{f_{\text{text}}(p(y_i))\}_{i=1}^N$:

$$\hat{y} = \underset{i \in \{1, \ldots, N\}}{\arg\max} \frac{f_{\text{img}}^\theta(x) \cdot f_{\text{text}}(p(y_i))}{\|f_{\text{img}}^\theta(x)\| \cdot \|f_{\text{text}}(p(y_i))\|},$$

where the numerator represents the dot product between image and text embeddings, and the denominator normalizes them using their $\ell_2$-norms.

## 3.3 Template-based Adversarial Fine-tuning

In this subsection, we present the learning objective of previous template-based adversarial fine-tuning methods and discuss their inherent limitations.

**Learning Objective.** Given an image $x \sim \mathcal{D}_{\mathcal{T}}$, its ground-truth label $y \in \{c_1, \ldots, c_N\}$, and templated text prompts $p(c_i)$ (e.g., "A photo of a {label}"), the attacker crafts an AE within an $\ell_p$-bounded perturbation set $\Delta = \{\delta : \|\delta\|_p \leq \epsilon\}$ by solving the following objective:

$$\delta^* = \underset{\delta \in \Delta}{\arg\max} \, \mathcal{L} \left( f_{\text{img}}^\theta(x + \delta), f_{\text{text}}(p(y)) \right),$$

where $\mathcal{L}$ is the cosine *dissimilarity* between image and text embeddings. In practice, this inner maximization is often approximated using *projected gradient descent* (PGD) (Madry et al., 2018), which iteratively updates the perturbation as follows:

$$\delta^{(t+1)} = \text{Proj}_\Delta \left( \delta^{(t)} + \alpha \cdot \text{sign} \left( \nabla_{\delta^{(t)}} \mathcal{L}(f_{\text{img}}^\theta(x + \delta^{(t)}), f_{\text{text}}(p(y))) \right) \right),$$

where $\delta^{(t)}$ is the perturbation at iteration $t$, $\alpha$ is the step size, and $\text{Proj}_\Delta(\cdot)$ denotes projection onto the $\ell_p$-ball of radius $\epsilon$. In contrast, the defender adversarially fine-tune $f_{\text{img}}^\theta$ to minimize the worst-case classification risk:

$$\min_\theta \; \mathbb{E}_{\mathcal{T} \sim \mathfrak{T}} \left[ \mathbb{E}_{(x, y) \sim \mathcal{D}_{\mathcal{T}}} \mathcal{L} \left( f_{\text{img}}^\theta(x + \delta^*), f_{\text{text}}(p(y)) \right) \right],$$

where $\theta$ refers to the parameters of CLIP's image encoder.

**Limitations.** Template-based adversarial fine-tuning relies on fixed prompts (e.g., "A photo of a {label}") to define class semantics in CLIP. While simple, this approach suffers from two main limitations: (1) AEs optimized by a single prompt often overfit to specific phrasings (e.g., "photo of a") rather than the class itself, failing to transfer to alternative expressions like "An image of a {label}"; (2) fixed templates fail to capture the rich attributes and contexts of real-world classes. For example, AEs generated for "dog" may not generalize to prompts like "a barking animal".

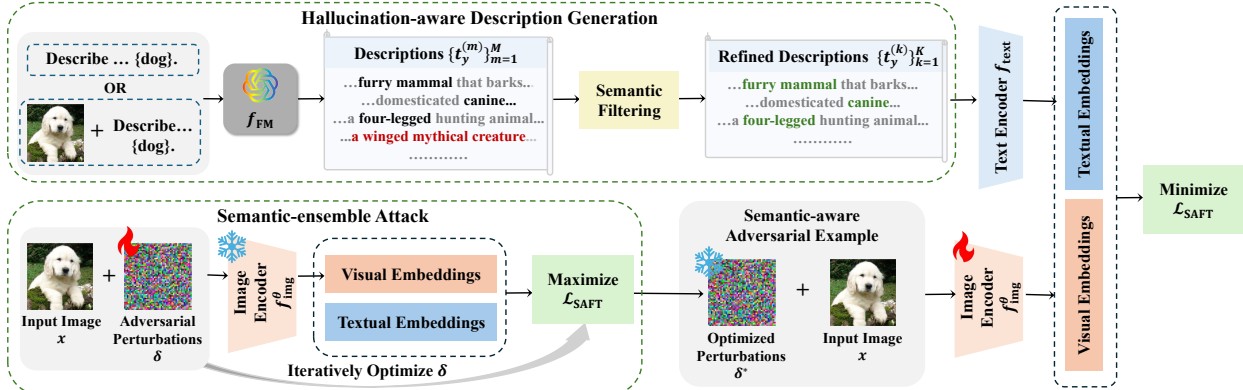

Figure 2: An overview of SAFT. In hallucination-aware description generation, a foundation model generates diverse textual descriptions for each class label, followed by the semantic filtering strategy to retain top-$K$ most relevant descriptions. These refined descriptions are then encoded by CLIP's text encoder. In semantic-ensemble attack, AEs are generated by maximizing misalignment between the visual embeddings and the average embeddings of refined descriptions. Finally, the image encoder is fine-tuned by minimizing this misalignment, aiming to learn linguistically invariant representations.

## 4 Semantic-aware Adversarial Fine-tuning

To address the limitations mentioned above, we propose a new framework called **S**emantic-aware **A**dversarial **F**ine-**T**uning (SAFT), which leverages a foundation model $f_{FM}$ (e.g., a LLM or a MLLM) to generate semantically enriched textual descriptions. Each AE is then crafted by minimizing the average similarity between the original image and the ensemble of these descriptions. In this section, we introduce key components of SAFT, which include hallucination-aware description generation, semantic-ensemble adversarial attack and the learning objective of SAFT. We provide the visual illustration in Figure 2 and the complete algorithm in Algorithm 1, respectively.

### 4.1 Hallucination-aware Description Generation

In this subsection, we present the hallucination-aware description generation, including the description generation method, its empirical realizations and the semantic filtering strategy.

**Description Generation Method.** Given a label space $\mathcal{Y}$, our goal is to construct a semantic mapping, where each class $y \in \mathcal{Y}$ is mapped to $M$ textual prompts $\{t_y^{(1)}, \ldots, t_y^{(M)}\}$ that capture diverse attributes, contexts, and synonyms associated with $y$. For instance, for the class "dog", a foundation model might generate semantic prompts such as "a furry mammal with four legs that barks", "a domesticated canine often kept as a pet", or "a loyal animal trained for hunting or companionship". These textual descriptions are generated using a foundation model $f_{FM}$, which we condition on class-specific instructions (e.g., "Describe the appearance of a {label}"). Formally, we define the generated descriptions $\mathcal{T}_y$ for a class $y$ as:

$$\mathcal{T}_y = \{t_y^{(1)}, \ldots, t_y^{(M)}\} = f_{FM}(y; \phi), \tag{1}$$

where $f_{FM}(y; \phi)$ denotes the output of the foundation model conditioned on label $y$ with generation hyperparameters $\phi$ (e.g., temperature for diversity). The resulting textual descriptions $\{t_y^{(m)}\}_{m=1}^{M}$ are subsequently encoded by CLIP's text encoder $f_{text}$ into $M$ embeddings. During adversarial fine-tuning, they offer more informative training signals by encouraging the image encoder to align perturbed inputs with the entire semantic manifold of $y$, rather than a single text template.

**Empirical Realizations.** In practice, we implement two different description generation methods by adopting recent studies (Pratt et al., 2023; Cai et al., 2025):

---

**Algorithm 1** Semantic-aware Adversarial Fine-tuning (SAFT)

---

**Require:** Pre-trained CLIP image encoder $f_{\text{img}}^{\theta}, f_{\text{text}}$; Class labels $\mathcal{Y} = \{y_1, \ldots, y_N\}$; Generation hyperparameters $\phi$; Training epochs $T$; Training dataset $\mathcal{D}_{\text{train}}$.
**Ensure:** Robust CLIP image encoder $f_{\text{img}}^{\theta^*}$.

1: **for** $y \in \mathcal{Y}$ **do**
2:   Generate $\{t_y^{(1)}, \ldots, t_y^{(M)}\} \sim f_{\text{FM}}(y; \phi)$
3:   Label embedding: $e_y \leftarrow f_{\text{text}}(y)$
4:   **for** $m = 1$ to $M$ **do**
5:     Compute $s^{(m)}$ by Eq. (2)
6:   **end for**
7:   Select $\{t_y^{(k)}\}_{k=1}^K$ by Eq. (3)
8:   Generate $\{f_{\text{text}}(t_y^{(k)})\}_{k=1}^K$
9: **end for**
10: **for** epoch $= 1$ to $T$ **do**
11:   **for** $(x_j, y_j) \sim \mathcal{D}_{\text{train}}$ **do**
12:     Compute the optimized $\delta_j^*$ by Eq. (4)
13:     Update $\theta$ by Eq. (5)
14:   **end for**
15: **end for**

---

1. **SAFT-L**. This method adopts CuPL (Pratt et al., 2023), which uses a LLM to generate class-specific textual descriptions. Given a class label, the LLM is prompted with a generic query *"What does a {label} look like?"* to produce texual descriptions.

2. **SAFT-M**. This method adopts (Cai et al., 2025), which propose to capture multiple common and unique features for each class. For each class, a set of representative images is provided along with prompts like *"Describe the unique appearance of a {label} compared to other objects."* and *"Describe the appearance of the object {label}."* to an MLLM. The MLLM then generates descriptions that contain descriptive and distinctive attributes.

We exclude WCA (Li et al., 2024) from our realizations primarily because it requires cropping each image into multiple patches, significantly increasing computational overhead. Additionally, its text description generation is aligned with CuPL (Pratt et al., 2023). Therefore, for SAFT-L, we only adopt CuPL to generate textual descriptions.

**Semantic Filtering Strategy.** While foundation models can generate diverse semantic prompts for class labels, they may occasionally produce *hallucinations:* descriptions that are semantically irrelevant or factually incorrect (e.g., describing a "dog" as "a mythical creature with wings"). To address this, we introduce a *semantic filtering* strategy that retains only descriptions closely aligned with the core semantics of each class. For a given class $y$, we first obtain a set of $M$ candidate prompts from the foundation model by Eq. (1), i.e., $\{t_y^{(1)}, \ldots, t_y^{(M)}\} = f_{\text{FM}}(y; \phi)$. We then compute the cosine similarity between each prompt $t_y^{(m)}$ and the ground-truth label $y$, both encoded by CLIP's text encoder $f_{\text{text}}$. The relevance score is defined as:

$$s^{(m)} = \frac{f_{\text{text}}(y) \cdot f_{\text{text}}(t_y^{(m)})}{\|f_{\text{text}}(y)\| \, \|f_{\text{text}}(t_y^{(m)})\|}, \ \forall m \in \{1, \ldots, M\}. \tag{2}$$

We sort the prompts by descending relevance scores:

$$\{t_y^{(m)}\} \rightarrow \text{Sort}\left(\{t_y^{(m)}\}, \{s^{(m)}\}\right), \ \text{s.t.} \ s^{(1)} \geq s^{(2)} \geq \cdots \geq s^{(M)}.$$

Finally, we retain the top-$K$ most relevant prompts to define the *refined descriptions* $\widetilde{\mathcal{T}}_y$:

$$\widetilde{\mathcal{T}}_y = \left\{t_y^{(k)}\right\}_{k=1}^K, \tag{3}$$

where each $t_y^{(k)}$ is among the top-$K$ prompts ranked by $s^{(m)}$. This filtering ensures that the final descriptions consists only of prompts that are semantically faithful to the class $y$. For example, for the class "dog", the foundation model may generate valid prompts like "a domesticated canine" alongside hallucinations like "a winged mythical creature." The hallucinated prompt typically receives a low relevance score (e.g., 0.2 vs. 0.8) and is filtered out. To this end, AEs are guided by *semantically meaningful* variations, avoiding noise introduced by model hallucinations. We further conduct the ablation study in Section 5.3.

### 4.2  Semantic-ensemble Adversarial Attack

Building on the hallucination-aware description generation, we further propose *semantic-ensemble adversarial attack*. Unlike conventional approaches that target a single prompt, we compute perturbations that maximize the *dissimilarity* between the perturbed image embedding $f_{\text{img}}^\theta(x+\delta)$ and the full set of textual embeddings $\{f_{\text{text}}(t_y^{(1)}), \ldots, f_{\text{text}}(t_y^{(M)})\}$. Formally, the optimized adversarial perturbation is obtained by solving:

$$\delta^* = \arg\max_{\delta \in \Delta} \mathcal{L}_{\text{SAFT}} \left( f_{\text{img}}^\theta(x+\delta), \{f_{\text{text}}(t_y^{(m)})\}_{m=1}^M \right), \tag{4}$$

where $\Delta = \{\delta : \|\delta\|_p \le \epsilon\}$, and $f_{\text{FM}}(y; \phi) = \{t_y^{(m)}\}_{m=1}^M$ denotes the textual descriptions for class $y$ generated by $f_{\text{FM}}$. The loss function $\mathcal{L}_{\text{SAFT}}$ measures the average cosine dissimilarity:

$$\mathcal{L}_{\text{SAFT}} \left( f_{\text{img}}^\theta(x+\delta), \{f_{\text{text}}(t_y^{(m)})\}_{m=1}^M \right) = -\frac{1}{M} \sum_{m=1}^M \frac{f_{\text{img}}^\theta(x+\delta) \cdot f_{\text{text}}(t_y^{(m)})}{\|f_{\text{img}}^\theta(x+\delta)\| \|f_{\text{text}}(t_y^{(m)})\|}.$$

This step produces stronger and more semantically enriched AEs, as it requires misalignment across multiple diverse prompts rather than a single template.

### 4.3  Learning Objective of SAFT

In general, SAFT formulates a bi-level optimization problem to enhance the adversarial robustness of CLIP's image encoder. The goal is to train the encoder parameters $\theta$ such that adversarially perturbed images remain well-aligned with the diverse textual semantics produced by a foundation model $f_{\text{FM}}$:

$$\min_\theta \mathbb{E}_{(x,y)} \left[ \mathcal{L}_{\text{SAFT}} \left( f_{\text{img}}^\theta(x+\delta^*), \{f_{\text{text}}(t_y^{(m)})\}_{m=1}^M \right) \right], \tag{5}$$

where $\delta^*$ is obtained by solving Eq. (4). Through this bi-level optimization, SAFT encourages the image encoder to learn representations that are robust not only to visual perturbations, but also invariant to linguistic variation across semantic prompts.

## 5  Experiments

### 5.1  Experiment Settings

**Datasets.** Following Yu et al. (2024), we evaluate all methods on 16 diverse datasets spanning general, fine-grained, and specialized classification tasks. These include standard benchmarks (e.g., CIFAR-10/100 (Krizhevsky et al., 2009), Tiny-ImageNet/ImageNet-1K (Deng et al., 2009), STL-10 (Coates et al., 2011), Caltech-101/256 (Griffin et al., 2007)), fine-grained datasets (e.g., Food101 (Bossard et al., 2014), Flowers102 (Nilsback & Zisserman, 2008), FGVC-Aircraft (Maji et al., 2013), StanfordCars (Krause et al., 2013), OxfordPets (Parkhi et al., 2012)), and domain-specific tasks (e.g., PCAM (Veeling et al., 2018), SUN397 (Xiao et al., 2010), EuroSAT (Helber et al., 2019) and DTD (Cimpoi et al., 2014)). Our main experiments use Tiny-ImageNet as the source dataset for adversarial fine-tuning, with the rest as unseen target tasks. We also include ImageNet-1K as an alternative source dataset in extended experiments (see Table 7 for more details).

**Baselines.** Following Yu et al. (2024), we compare SAFT with five representative baselines: (i) CLIP (Radford et al., 2021) fine-tuned on clean data; (ii) TeCoA (Mao et al., 2023), which applies adversarial training with fixed zero-shot prompts; (iii) FARE (Schlarmann et al., 2024), which adds unsupervised objectives to

Table 1: Clean and robust accuracy (%) against PGD-100 ($\epsilon = 1/255$) of different methods across 16 datasets. Tiny-ImageNet is the source dataset and the others are zero-shot datasets. The best accuracy is highlighted in **bold** and the second-best accuracy is underlined. "Zero-shot Average" refers to the averaged clean/robust accuracy across 15 datasets for evaluating zero-shot classification. We report standard deviations and averaged results of SAFT-L and SAFT-M for three runs.

| | Method | Tiny-ImageNet | CIFAR-10 | CIFAR-100 | Food101 | STL-10 | OxfordPets | Flowers102 | DTD | EuroSAT | FGVC-Aircraft | Caltech-101 | Caltech-256 | StanfordCars | PCAM | ImageNet-1K | SUN397 | Zero-shot Average |
|---|---|---|---|---|---|---|---|---|---|---|---|---|---|---|---|---|---|---|
| | | Source | | | | | | | | | | | | | | | | |
| **Robust** | CLIP | 13.55 | 19.92 | 4.94 | 0.64 | 40.00 | 2.40 | 0.68 | 2.66 | 0.05 | 0.03 | 14.95 | 9.69 | 0.09 | 1.32 | 1.08 | 0.82 | 6.62 |
| | TeCoA | 31.29 | 31.56 | 16.16 | 14.35 | 65.93 | 34.39 | 19.09 | 16.39 | 10.19 | 1.87 | 52.92 | 40.62 | 8.74 | 45.57 | 15.97 | 16.23 | 26.00 |
| | FARE | 23.88 | 21.25 | 10.72 | 10.97 | 59.59 | 24.56 | 15.48 | 10.96 | 0.14 | 0.84 | 45.96 | 34.35 | 4.38 | 10.17 | 10.54 | 8.30 | 17.88 |
| | PMG-AFT | 32.46 | 50.58 | 26.42 | 16.59 | 68.79 | 35.57 | 15.69 | 12.85 | 9.33 | 1.85 | 53.33 | 36.30 | 5.54 | **48.92** | 15.02 | 16.37 | 27.54 |
| | TGA-ZSR | 47.87 | 42.61 | 22.27 | 20.75 | 75.47 | 42.32 | 25.46 | 18.45 | 10.84 | 3.51 | 60.91 | 50.43 | 12.91 | 42.10 | 21.11 | 23.29 | 31.50 |
| | SAFT-L | **53.27** | **52.26** | **28.34** | 25.67 | **78.97** | **46.07** | **30.22** | **20.74** | **12.42** | **4.53** | **64.67** | **54.27** | **15.86** | 47.09 | **22.74** | **26.33** | **35.35** |
| | | (± 1.56) | (± 1.28) | (± 1.09) | (± 1.55) | (± 0.71) | (± 3.81) | (± 1.78) | (± 0.12) | (± 1.65) | (± 0.29) | (± 1.22) | (± 1.02) | (± 0.34) | (± 2.59) | (± 0.40) | (± 0.38) | (± 1.04) |
| | SAFT-M | 50.79 | 45.32 | 25.29 | **25.84** | 78.48 | 43.29 | 29.48 | 19.31 | 11.95 | 4.02 | 63.36 | 53.08 | 14.40 | 46.80 | 22.43 | 23.62 | 33.78 |
| | | (± 1.43) | (± 1.17) | (± 0.98) | (± 1.61) | (± 0.64) | (± 3.45) | (± 1.69) | (± 0.21) | (± 1.49) | (± 0.36) | (± 1.11) | (± 0.89) | (± 0.27) | (± 2.74) | (± 0.48) | (± 0.42) | (± 1.15) |
| **Clean** | CLIP | **79.04** | 84.55 | 54.25 | 47.10 | 93.78 | 80.98 | 46.43 | 30.32 | 24.39 | 9.30 | 78.69 | 70.81 | 31.15 | 47.89 | 44.40 | 46.80 | 52.72 |
| | TeCoA | 54.59 | 66.78 | 34.37 | 33.60 | 85.90 | 61.81 | 34.06 | 24.79 | 18.83 | 6.42 | 70.67 | 60.08 | 22.23 | **52.32** | 31.85 | 34.53 | 43.08 |
| | FARE | 77.54 | **87.58** | **62.80** | **70.02** | **94.33** | **81.47** | **57.10** | **36.33** | 22.69 | **14.19** | **84.04** | **77.50** | **44.35** | 46.07 | **51.78** | 49.91 | 58.68* |
| | PMG-AFT | 47.36 | 71.20 | 40.53 | 28.77 | 81.83 | 54.28 | 24.14 | 18.28 | 13.12 | 3.84 | 64.73 | 48.49 | 10.77 | 49.93 | 23.72 | 27.58 | 37.42 |
| | TGA-ZSR | 76.84 | 85.90 | 57.20 | 56.46 | 92.92 | 75.47 | 47.51 | 29.94 | 24.60 | 11.98 | 80.02 | 73.82 | 36.32 | 49.91 | 46.50 | 50.16 | 54.58 |
| | SAFT-L | 73.47 | 85.81 | 57.26 | 58.69 | 93.30 | 77.90 | 48.96 | 30.05 | 25.83 | 13.39 | 79.89 | 74.39 | 35.18 | 49.86 | 46.61 | 51.91 | 55.27 |
| | | (± 0.38) | (± 0.76) | (± 0.87) | (± 0.49) | (± 0.26) | (± 1.07) | (± 3.06) | (± 0.22) | (± 5.69) | (± 1.23) | (± 1.22) | (± 0.19) | (± 0.58) | (± 0.59) | (± 0.72) | (± 0.29) | (± 0.52) |
| | SAFT-M | 74.88 | 86.49 | 60.94 | 58.43 | 94.63 | 75.09 | 49.13 | 30.64 | 29.56 | 13.86 | 81.35 | 75.26 | 34.41 | 49.99 | 47.30 | 53.08 | 56.01 |
| | | (± 0.41) | (± 0.84) | (± 0.91) | (± 0.47) | (± 0.33) | (± 1.15) | (± 2.97) | (± 0.28) | (± 5.42) | (± 1.09) | (± 1.31) | (± 0.25) | (± 0.64) | (± 0.53) | (± 0.80) | (± 0.36) | (± 0.61) |

*Although FARE achieves the highest clean accuracy, SAFT-L surpasses it by 17.46% and SAFT-M by 15.90% in averaged zero-shot robustness.

strengthen adversarial examples; (iv) PMG-AFT (Wang et al., 2024a), which uses auxiliary branches to align clean and adversarial representations and mitigate overfitting; and (v) TGA-ZSR (Yu et al., 2024), which leverages text-guided attention and achieves SOTA performance on zero-shot robustness benchmarks.

**Implementation Details.** We adopt TGA-ZSR (Yu et al., 2024) as the default loss function to further push the upper bound of CLIP's zero-shot accuracy-robustness trade-off by building on its strong foundation. We use refined descriptions with top-5 relevance scores (see Section 5.3). To ensure fair comparison, we follow Yu et al. (2024) in using ViT-B/32 as the backbone and SGD optimizer with a learning rate of 1e-4, momentum 0.9, weight decay 0, and batch size 128. During training, AEs are generated via $\ell_\infty$-norm PGD-2 (Madry et al., 2018) with $\epsilon = 1/255$. For evaluation, we use mainly use PGD, AutoAttack (Croce & Hein, 2020), and C&W (Carlini & Wagner, 2017) to evaluate zero-shot robustness. For experiments fine-tuned on Tiny-ImageNet, all models are fine-tuned for 10 epochs. Due to limited computational resources, for experiments fine-tuned on ImageNet-1K, all models are fine-tuned for only 1 epoch.

**Hyperparameters for Generating Descriptions.** For SAFT-L, we adopt (Pratt et al., 2023), which uses LLMs to generate class-specific textual descriptions. All descriptions are generated by the prompt *"What does a {label} look like"*. These textual descriptions are publicly available at: https://github.com/sarahpratt/CuPL/tree/main. For SAFT-M, we adopt (Cai et al., 2025), which propose to capture multiple common and unique features for each class. For each label, we provide 5 images that are randomly sampled from the training data with *"This is a photo of {label}"* and additional prompts *"Describe the unique appearance of a {label} compared to other objects. Use one short sentence"* and *"Describe the appearance of the object {label}. Use one short sentence."*. The MLLM we use is GPT-4o-mini (OpenAI, 2023). We set the temperature to be 0.99, max tokens to be 100 and number of responses for each prompt to be 5.

## 5.2 Evaluation of Zero-shot Classification

**Result Analysis on Tiny-ImageNet.** As shown in Table 1, both SAFT-L and SAFT-M significantly improves zero-shot robustness against PGD-100, outperforming the previous SOTA (TGA-ZSR) by 3.85% and 2.28% on average across 15 target datasets and by 5.40% and 2.92% on Tiny-ImageNet, respectively. Notably, SAFT-L consistently performs better across most of the remaining datasets, validating the benefit of semantically enriched text supervision in adversarial fine-tuning. In terms of clean accuracy, SAFT-L and SAFT-M rank second and third overall, achieving 55.27% and 56.01% average clean accuracy, respectively.

Table 2: Robust accuracy (%) of different methods against AutoAttack ($\epsilon = 1/255$) and C&W attack ($\epsilon = 1/255$) across 16 datasets. Tiny-ImageNet is the source dataset and the others are zero-shot datasets. The best accuracy is highlighted in **bold** and the second-best accuracy is underlined.

| Method | Source Tiny-ImageNet | CIFAR-10 | CIFAR-100 | Food101 | STL-10 | OxfordPets | Flowers102 | DTD | EuroSAT | FGVC-Aircraft | Caltech-101 | Caltech-256 | StanfordCars | PCAM | ImageNet-1K | SUN397 | Zero-shot Average |
|---|---|---|---|---|---|---|---|---|---|---|---|---|---|---|---|---|---|
| | | | | | | AutoAttack ($\epsilon = 1/255$) | | | | | | | | | | | |
| TeCoA | 27.21 | 29.81 | 14.23 | 11.64 | 66.20 | 32.46 | 16.91 | 13.09 | 5.77 | 1.29 | 51.55 | 38.58 | 6.77 | 32.71 | 16.21 | 17.41 | 23.64 |
| PMG-AFT | 29.99 | **48.66** | 23.64 | 14.32 | 67.59 | 32.41 | 13.86 | 11.65 | 8.81 | 0.99 | 51.26 | 40.25 | 9.59 | 38.53 | 19.17 | 19.63 | 26.69 |
| TGA-ZSR | 48.95 | 40.28 | 22.33 | 15.03 | 71.90 | **39.49** | 21.81 | 16.38 | 11.27 | 2.31 | 57.75 | 45.41 | 10.20 | 40.86 | 19.20 | 19.11 | 28.89 |
| SAFT-L | **50.40** | 42.88 | **24.54** | **15.12** | **73.93** | 36.11 | **22.65** | **17.55** | **12.91** | **2.79** | **59.64** | **45.64** | **12.19** | **43.96** | **20.22** | **20.81** | **30.06** |
| | | | | | | C&W ($\epsilon = 1/255$) | | | | | | | | | | | |
| TeCoA | 28.25 | 31.11 | 14.85 | 12.01 | 66.65 | 33.96 | 17.29 | 13.09 | 7.26 | 1.50 | 39.34 | 52.66 | 7.60 | 32.06 | 14.24 | 15.26 | 23.93 |
| PMG-AFT | 30.30 | 49.11 | 23.87 | 14.36 | 67.64 | 32.65 | 13.90 | 11.44 | 8.76 | 1.05 | 51.55 | 33.81 | 4.69 | **48.33** | 12.92 | 13.33 | 25.83 |
| TGA-ZSR | 44.53 | 36.42 | 19.19 | 19.56 | 74.00 | 42.52 | 22.87 | 16.49 | 9.90 | 2.91 | 59.88 | 49.73 | 12.34 | 41.48 | 19.89 | 21.95 | 29.94 |
| SAFT-L | **58.21** | **57.73** | **32.04** | **30.50** | **82.19** | **50.40** | **31.52** | **20.96** | **13.19** | **3.93** | **68.23** | **57.68** | **17.14** | 42.93 | **24.53** | **28.73** | **37.45** |

Table 3: Ablation study on semantic filtering. We report clean and robust accuracy (%) against PGD-100 ($\epsilon = 1/255$) of SAFT-L across 16 datasets. Tiny-ImageNet is the source dataset and the others are zero-shot datasets. The best accuracy is highlighted in **bold**.

| | Semantic Filtering | Source Tiny-ImageNet | CIFAR-10 | CIFAR-100 | Food101 | STL-10 | OxfordPets | Flowers102 | DTD | EuroSAT | FGVC-Aircraft | Caltech-101 | Caltech-256 | StanfordCars | PCAM | ImageNet-1K | SUN397 | Zero-shot Average |
|---|---|---|---|---|---|---|---|---|---|---|---|---|---|---|---|---|---|---|
| Robust | ✗ | 53.15 | 51.88 | 27.25 | 25.26 | 80.20 | 44.69 | 25.87 | 18.40 | 10.96 | 4.02 | 61.53 | 51.29 | 13.09 | 46.09 | 20.69 | 23.36 | 33.64 |
| | ✔ | **53.27** | **52.26** | **28.34** | **25.67** | 78.97 | **46.07** | **30.22** | **20.74** | **12.42** | **4.53** | **64.67** | **54.27** | **15.86** | **47.09** | **22.74** | **26.33** | **35.35** |
| Clean | ✗ | 73.03 | **86.52** | 56.05 | **59.02** | **94.04** | 77.05 | 45.32 | 29.40 | 25.76 | 12.19 | **80.72** | 73.88 | 32.40 | 49.73 | 44.86 | 50.44 | 54.49 |
| | ✔ | **73.47** | 85.81 | **57.26** | 58.69 | 93.30 | **77.90** | **48.96** | **30.05** | **25.83** | **13.39** | 79.89 | **74.39** | **35.18** | **49.86** | **46.61** | **51.91** | **55.27** |

Although FARE achieves the highest clean accuracy, SAFT-L surpasses it by 17.46% in average zero-shot robustness, and SAFT-M by 15.90%. These results demonstrate that SAFT-based methods can notably improve the accuracy-robustness trade-off.

**Result Analysis on ImageNet-1K.** As shown in Table 7, when scaling to ImageNet-1K, SAFT-M exhibits notably improved performance in zero-shot clean accuracy, achieving the highest overall score of 63.69% across 15 target datasets. This demonstrates the scalability of SAFT-based methods under larger-scale pre-training. In terms of robustness, while PMG-AFT achieves the highest zero-shot robust accuracy, it comes at the cost of significantly lower clean accuracy. In contrast, both SAFT-M and SAFT-L attain comparable robustness while outperforming PMG-AFT by 8.70% and 8.15% in clean accuracy, respectively. This clearly highlights the advantage of SAFT-based methods in achieving a more favorable accuracy-robustness trade-off when fine-tuned on large-scale datasets.

### 5.3 Ablation Studies

**Ablation Study on $K$.** We investigate how the number of selected textual descriptions $K$ in Eq. (3) affects the performance of SAFT-L against PGD-100 on four datasets, including one source dataset (i.e., Tiny-ImageNet) and three zero-shot datasets (i.e., CIFAR-10, CIFAR-100 and STL-10) in Appendix A.2. We find that more textual descriptions per class do not necessarily imply better performance. Specifically, when $K = 5$, SAFT-L can achieve the best robustness-accuracy trade-off on average (see Table 8). Therefore, in this paper, *we use $K = 5$ for all the experiments.*

**Ablation Study on Unseen Attacks.** We also evaluate transferability to unseen attacks, using AutoAttack and C&W across 16 datasets in Table 2. We compare SAFT-L with the top three baselines on zero-shot

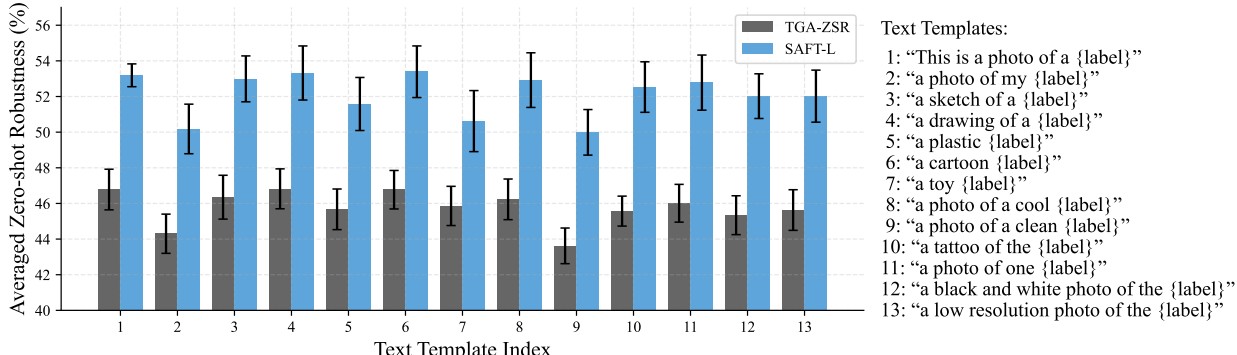

Figure 3: Transferability to different text templates. We compare SAFT-L and TGA-ZSR across 13 text templates, using Tiny-ImageNet as the source dataset and PGD-100 as the evaluation method. We report the standard deviations and averaged zero-shot robustness (%) on CIFAR-10, CIFAR-100, and STL-10 for three runs. SAFT-L *consistently* outperforms TGA-ZSR in averaged zero-shot robust accuracy across all templates. We provide full experimental results in Appendix A.4.

Table 4: Clean and robust accuracy (%) against PGD-100 ($\epsilon = 1/255$) of different methods across 16 datasets using CLIP-L/14. Tiny-ImageNet is the source dataset and the others are zero-shot datasets.

| | | Source | Datasets for Evaluating Zero-shot Classification Accuracies | | | | | | | | | | | | | | |
|---|---|---|---|---|---|---|---|---|---|---|---|---|---|---|---|---|---|
| | Method | Tiny-ImageNet | CIFAR-10 | CIFAR-100 | Food101 | STL-10 | OxfordPets | Flowers102 | DTD | EuroSAT | FGVC-Aircraft | Caltech-101 | Caltech-256 | StanfordCars | PCAM | ImageNet-1K | SUN397 | Zero-shot Average |
| Clean | FARE | 68.23 | 89.19 | 57.77 | **72.12** | **97.01** | **87.08** | 63.26 | 35.80 | 25.32 | **20.61** | **86.71** | 81.89 | **56.19** | 50.89 | 55.15 | 49.33 | 61.89 |
| | SAFT-L | **80.39** | **89.49** | **61.84** | 66.24 | 95.64 | 86.64 | **66.24** | **39.15** | **32.13** | 18.06 | 86.70 | **82.04** | 52.03 | **57.17** | **57.43** | **56.02** | **63.12** |
| Robust | FARE | 42.20 | 45.82 | 30.73 | 36.74 | 80.18 | 70.73 | 46.85 | 24.68 | 12.07 | 11.31 | 75.50 | 64.35 | 36.31 | 49.42 | 35.09 | 32.89 | 43.51 |
| | SAFT-L | **68.06** | **66.60** | **45.50** | **51.38** | **95.64** | **86.64** | **66.24** | **37.00** | **23.22** | **16.59** | **79.07** | **72.15** | **43.86** | **50.52** | **50.70** | **50.93** | **53.52** |

robustness (i.e., TGA-ZSR, PMG-AFT and TeCoA). Results show that SAFT-L consistently improves zero-shot robust accuracy, achieving average gains of 1.17% on AutoAttack and 7.51% on C&W attack.

**Ablation Study on Semantic Filtering.** We investigate the effect of semantic filtering in our method by comparing performance with and without this step under PGD-100 attacks in Table 3. Incorporating semantic filtering yields an average improvement of 1.71% in zero-shot robust accuracy. This suggests that the quality of textual descriptions is key to our method.

**Ablation Study on LLMs.** We conduct an ablation study using Qwen3-4b-instruct (Yang et al., 2025), Qwen3-4b (Yang et al., 2025), Llama-3.2-1b-instruct (Grattafiori et al., 2024) and Llama-3.2-3b-instruct (Grattafiori et al., 2024) to generate descriptions, respectively. Overall, as shown in Appendix A.3, instruction-tuned models with larger size can generate better descriptions. However, we want to highlight that, in our framework, the quality of semantic descriptions is the fundamental factor that drives robustness improvements. LLMs serve as a practical and scalable implementation for generating such descriptions, but they are not an essential component of the method itself. Any mechanism capable of producing semantically accurate and conceptually rich descriptions could, in principle, be used in place of an LLM.

### 5.4 More Empirical Analysis

**Transferability to Different Text Templates.** Prior methods use a single template (e.g. "This is a photo of a {label}") to define class semantics, which limits the robustness to semantic variations. We investigate the transferability of SAFT-L and TGA-ZSR on 13 templates[1], including 1 seen (i.e., "This is a photo of a {label}") and 12 unseen variants, across 3 target datasets (i.e., CIFAR-10, CIFAR-100, and STL-10). As shown in Figure 3, SAFT-L consistently outperforms TGA-ZSR in averaged zero-shot robust accuracy across

---

[1]Prompt templates are randomly selected from https://github.com/chs20/RobustVLM/blob/main/CLIP_eval/zeroshot-templates.json.

Table 5: Image retrieval recall (IRR) (%) and text retrieval recall (TRR) (%) of different methods on COCO, Flickr8k, and Flickr30k against AutoAttack ($\epsilon = 1/255$).

| Method | COCO | | Flickr8k | | Flickr30k | |
|---|---|---|---|---|---|---|
| | IRR | TRR | IRR | TRR | IRR | TRR |
| TGA-ZSR | 34.27 | 41.64 | 57.40 | 66.80 | 61.14 | 69.40 |
| SAFT-L | **35.36** | **43.32** | **58.86** | **68.60** | **63.86** | **72.10** |

Table 6: Comparison of training memory usage and training time of different methods.

| Methods | GPU | Batch Size | Training Memory Usage | Training Time (Per Epoch / Batch) |
|---|---|---|---|---|
| TeCoA | | | 13977Mb | 801s / 1.03s |
| PMG-AFT | 1 * NVIDIA A100 | 128 | 18423Mb | 660s / 0.84s |
| TGA-ZSR | | | 21253Mb | 786s / 1.01s |
| SAFT-L ($K = 5$) | | | 38839Mb | 1098s / 1.41s |

both seen and unseen templates. We provide full experimental results in Appendix A.4. Specifically, SAFT-L improves averaged zero-shot robust accuracy by 6.41% on the training template and up to 6.96% on unseen variants such as "a tattoo of the {label}". These results highlight SAFT's ability to capture semantic diversity beyond fixed linguistic patterns.

**Adaptability to Larger Perturbation Budget.** We further evaluate our method under a larger perturbation budget in Appendix A.5. This evaluates whether models trained on weaker AEs generalize to stronger attacks. In Table 11, under $\epsilon = 4/255$, SAFT-L still outperforms baseline methods by at least 1.52%.

**Scalability to CLIP-L/14.** We investigate the scalability of SAFT-L on CLIP-L/14 in Table 4. Notably, when scaled to CLIP-L/14, SAFT-L outperforms FARE by approximately 2% in clean accuracy and 10% in robust accuracy, demonstrating its strong scalability to larger VLMs. This result is particularly surprising, as SAFT-L slightly lagged behind FARE in clean accuracy when using CLIP-B/32, suggesting that larger VLMs better leverage the semantic richness during the fine-tuning.

**Applicability to Image-text Retrieval Task.** To demonstrate SAFT's applicability beyond classification, we conduct an additional experiment on the image-text retrieval task in Table 5. Specifically, we replace the original CLIP image encoder with the adversarially fine-tuned image encoder from SAFT-L and TGA-ZSR. We compare the image retrieval recall and text retrieval recall on 3 benchmark image-text retrieval datasets: COCO (Lin et al., 2014), Flickr8k (Hodosh et al., 2013), and Flickr30k (Plummer et al., 2015). Our method consistently outperforms TGA-ZSR by a notable margin.

**Generalizability to Out-of-domain Datasets.** To evaluate whether the robustness gains of SAFT extend beyond ImageNet-like visual semantics, we further test the models on two genuinely out-of-domain datasets: TU-Berlin sketch dataset (Eitz et al., 2012) and ImageNet-R (Hendrycks et al., 2020). They differ substantially from ImageNet-like natural images. Sketch consists of hand-drawn line sketches without texture or color information, while ImageNet-R contains artistic renditions such as cartoons and paintings that preserve high-level semantics but significantly alter visual appearance. As shown in Appendix A.6, SAFT-L consistently outperforms TGA-ZSR by a notable margin.

### 5.5 Compute Resources

We compare the memory usage and training time of SAFT-L[2] with baseline methods in Table 6. Inference time is similar across all methods and thus omitted. Introducing multiple semantic descriptions increases memory and training overhead. For example, SAFT requires 0.4 seconds more per batch than TGA-ZSR. However, given the trade-off between computational cost and the performance of SAFT, it is worthwhile to introduce semantically enriched text during fine-tuning.

---

[2]SAFT-M consumes exactly the same memory usage and training time as SAFT-L.

## 6 Limitations

**Dependence on the Quality of Foundation Models.** SAFT relies on foundation models to generate semantic descriptions for each class. The effectiveness of this process depends on the relevance, faithfulness, and diversity of the generated descriptions. In cases where the foundation model produces hallucinated or semantically ambiguous prompts, the resulting descriptions may be suboptimal. To mitigate this issue, we propose a *semantic filtering strategy* to filter out descriptions with low relevance scores.

**Extra Computational Cost.** The integration of an ensemble of textual descriptions will inevitably bring some extra cost. Luckily, we find that this process is lightweight, making SAFT computationally feasible compared to existing adversarial fine-tuning methods.

## 7 Conclusion

In this paper, we find that AEs generated using cosine similarity may fail to fool CLIP when the similarity metric is replaced with semantically enriched alternatives, making the image encoder fine-tuned with these AEs less robust. To address this problem, we propose $\boldsymbol{S}$emantic-aware $\boldsymbol{A}$dversarial $\boldsymbol{F}$ine-$\boldsymbol{T}$uning (SAFT), a new framework that generates semantic-aware AEs by incorporating hallucination-aware textual descriptions during the fine-tuning. Extensive experiments show that SAFT notably improves zero-shot adversarial robustness across 16 datasets compared to current SOTA methods and can generalize well to unseen templates. In general, we hope this simple yet effective framework could open up a new perspective in the adversarial fine-tuning of CLIP and lay the groundwork for future methods that account for richer text supervision.

## Impact Statement

This study on adversarial fine-tuning for CLIP raises important ethical considerations that we have carefully addressed. We have taken steps to ensure our method is fair. We use widely accepted public benchmark datasets to ensure comparability of our results. Our evaluation encompasses a wide range of attack types and strengths to provide a comprehensive assessment. We have also carefully considered the broader impacts of our work. The proposed adversarial fine-tuning algorithm contributes to the development of more robust machine learning models, potentially improving the reliability of AI systems in various applications. We will actively engage with the community to promote responsible development and use of adversarial fine-tuning.

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

# A   Additional Experiments

## A.1   Experiment on ImageNet-1K

Table 7: Clean and robust accuracy (%) against PGD-100 ($\epsilon = 1/255$) of different methods across 16 datasets. ImageNet is the source dataset and the others are zero-shot datasets. The best accuracy is highlighted in **bold** and the second-best accuracy is underlined. "Zero-shot Average" refers to the averaged clean/robust accuracy across 15 zero-shot datasets.

| | Method | ImageNet-1K (Source) | CIFAR-10 | CIFAR-100 | Food101 | STL-10 | OxfordPets | Flowers102 | DTD | EuroSAT | FGVC-Aircraft | Caltech-101 | Caltech-256 | StanfordCars | PCAM | Tiny-ImageNet | SUN397 | Zero-shot Average |
|---|---|---|---|---|---|---|---|---|---|---|---|---|---|---|---|---|---|---|
| Clean | CLIP | **67.41** | 88.18 | 62.81 | 73.79 | **97.09** | 84.71 | 56.24 | 39.15 | **40.99** | 15.81 | 83.98 | **83.40** | 42.81 | 41.45 | 66.52 | 61.16 | 62.54 |
| | TeCoA | 52.38 | 72.02 | 41.27 | 56.30 | 92.25 | 77.21 | 48.48 | 33.99 | 20.07 | 15.21 | 82.61 | 77.56 | 34.17 | 43.80 | 53.52 | 55.17 | 53.58 |
| | FARE | 54.90 | 77.81 | 52.33 | 70.16 | 94.29 | 84.57 | 57.55 | 35.90 | 21.99 | 16.02 | 83.37 | 78.42 | **47.33** | 47.27 | 55.88 | 51.52 | 58.29 |
| | PMG-AFT | 49.39 | 80.43 | 49.66 | 65.73 | 93.99 | 75.22 | 48.30 | 32.55 | 23.39 | 12.06 | 83.60 | 74.69 | 30.31 | 48.21 | 56.24 | 54.99 | 54.99 |
| | TGA-ZSR | 63.44 | 88.25 | 57.65 | 73.94 | 95.33 | 84.06 | 60.43 | 38.03 | 39.54 | **19.08** | 81.66 | 81.73 | 45.73 | 46.34 | 64.45 | 60.23 | 62.43 |
| | SAFT-L | 63.37 | 88.38 | 60.52 | **74.33** | 95.94 | **85.47** | **61.29** | 40.21 | 34.93 | 17.85 | 82.95 | 82.79 | 46.50 | 47.29 | 65.75 | **62.92** | 63.14 |
| | SAFT-M | 64.04 | **89.13** | **63.03** | 70.66 | 95.83 | 83.57 | 58.95 | **40.85** | 36.35 | 17.19 | **85.01** | 82.40 | 45.00 | **57.86** | **69.22** | 60.29 | **63.69** |
| Robust | CLIP | 3.78 | 7.62 | 2.94 | 3.45 | 33.46 | 12.02 | 0.89 | 2.18 | 0.04 | 0.06 | 30.66 | 21.58 | 0.20 | 0.49 | 5.05 | 1.16 | 7.85 |
| | TeCoA | 28.55 | 33.01 | 17.38 | 25.19 | 70.35 | 49.88 | 27.76 | 22.66 | 12.23 | **5.22** | 67.70 | 57.18 | 11.29 | 21.41 | 21.94 | **28.91** | 31.29 |
| | FARE | 25.97 | 37.13 | **19.37** | 25.56 | 72.85 | 72.73 | 27.73 | 20.96 | 11.39 | 3.90 | 63.02 | 53.00 | 14.04 | 20.05 | 21.27 | | 30.86 |
| | PMG-AFT | 29.88 | **40.64** | 17.77 | 24.44 | **81.08** | 51.16 | 27.83 | **23.51** | **14.41** | 4.62 | 69.07 | 57.22 | 10.28 | 23.27 | 22.16 | 27.12 | **32.78** |
| | TGA-ZSR | 30.30 | 26.18 | 12.41 | 24.41 | 71.86 | 52.41 | 27.19 | 20.74 | 7.36 | 4.62 | 67.36 | 55.37 | 12.35 | 13.16 | 15.12 | 22.40 | 28.95 |
| | SAFT-L | 30.82 | 28.41 | 13.56 | **26.19** | 73.76 | **56.12** | **29.84** | 22.55 | 12.76 | 3.75 | 69.67 | 57.70 | 13.47 | 11.17 | 18.37 | 24.97 | 30.82 |
| | SAFT-M | **32.42** | 30.36 | 16.22 | 24.13 | 71.53 | 53.53 | 26.43 | 21.91 | 12.05 | 4.99 | **69.97** | **59.92** | **14.31** | 21.84 | **22.66** | 25.23 | 31.67 |

## A.2   Ablation Study on Number of Descriptions

Table 8: Ablation study on number of descriptions per class. We use $M$ to represent the number of descriptions selected for each class. Tiny-ImageNet is the source dataset and CIFAR-10, CIFAR-100, and STL-10 are zero-shot datasets. The best accuracy is highlighted in **bold** and the second-best accuracy is underlined. We report the averaged results of three runs.

| No. of Descriptions | Tiny-ImageNet (Source) Clean | Tiny-ImageNet (Source) Robust | CIFAR-10 Clean | CIFAR-10 Robust | CIFAR-100 Clean | CIFAR-100 Robust | STL-10 Clean | STL-10 Robust | Average Clean | Average Robust |
|---|---|---|---|---|---|---|---|---|---|---|
| $K = 1$ | 72.00 | 49.70 | 84.92 | 47.89 | 56.26 | 25.77 | 93.34 | 77.68 | 76.63 | 50.26 |
| $K = 2$ | 73.18 | 50.21 | 84.85 | 49.10 | 56.29 | 26.10 | 92.85 | 77.65 | 76.80 | 50.77 |
| $K = 3$ | 73.17 | 43.04 | 84.86 | 39.21 | 56.66 | 19.18 | 92.83 | 74.05 | 76.88 | 43.87 |
| $K = 4$ | 73.34 | 44.08 | 85.11 | 42.24 | 56.98 | 20.95 | 92.73 | 74.15 | 77.04 | 45.35 |
| $K = 5$ | 73.47 | **53.27** | **85.81** | **52.26** | 57.26 | **28.34** | 93.30 | **78.97** | 77.46 | **53.21** |
| $K = 6$ | **73.83** | 48.00 | 85.38 | 45.42 | 57.05 | 23.48 | 93.53 | 77.58 | 77.45 | 48.62 |
| $K = 7$ | 73.51 | 44.11 | 84.86 | 41.27 | 57.77 | 19.37 | 93.21 | 75.90 | 77.34 | 45.16 |
| $K = 8$ | 73.17 | 47.90 | 85.22 | 45.55 | 57.39 | 22.49 | 92.81 | 76.63 | 77.15 | 48.14 |
| $K = 9$ | 73.41 | 45.33 | 85.15 | 43.66 | 57.60 | 21.15 | 93.50 | 75.74 | 77.41 | 46.47 |
| $K = 10$ | 73.66 | 43.86 | 85.41 | 41.30 | **58.52** | 20.63 | **93.56** | 74.67 | **77.79** | 45.11 |

## A.3 Ablation Study on LLMs

Table 9: Ablation study on LLMs of varying quality and scale. We report the best in **bold**.

| | LLM | Source | Zero-shot Datasets | | |
|---|---|---|---|---|---|
| | | Tiny-ImageNet | CIFAR-10 | CIFAR-100 | STL10 |
| Clean | Qwen3-4B-instruct | 73.96 | 85.45 | 55.46 | 93.58 |
| | Qwen3-4B | 73.22 | 86.35 | **56.98** | **93.73** |
| | Llama-3.2-1B-instruct | 74.00 | **86.39** | 56.42 | 93.43 |
| | Llama-3.2-3B-instruct | **74.05** | 86.25 | 56.28 | 92.31 |
| Robust | Qwen3-4B-instruct | **53.45** | **51.91** | **27.79** | **78.91** |
| | Qwen3-4B | 47.10 | 44.96 | 23.40 | 76.08 |
| | Llama-3.2-1B-instruct | 49.08 | 46.15 | 23.98 | 77.14 |
| | Llama-3.2-3B-instruct | 50.14 | 48.34 | 26.40 | 78.34 |

## A.4 Transferability to Different Text Templates

Table 10: Ablation study on text templates. We compare clean and robust accuracy (%) against PGD-100 ($\epsilon = 1/255$) between TGA-ZSR and our method across 4 datasets. Tiny-ImageNet is the source dataset and the others are zero-shot datasets. 13 different text templates are used for evaluation, which include 1 seen template ("This is a photo of a {label}") and 12 unseen templates. The best accuracy is highlighted in **bold**. The performance improvements and degradation are reported in green and red, respectively. We report the averaged results of three runs.

| Type | Template | Method | Source Tiny-ImageNet Clean | Robust | CIFAR-10 Clean | Robust | CIFAR-100 Clean | Robust | STL-10 Clean | Robust | Zero-shot Average Clean | Robust |
|---|---|---|---|---|---|---|---|---|---|---|---|---|
| Seen | "This is a photo of a {label}" | TGA-ZSR | **76.84** | 47.87 | **85.90** | 42.61 | 57.20 | 22.27 | 92.92 | 75.47 | 78.67 | 46.78 |
| | | SAFT-L | 73.47 | **53.27** | 85.81 | **52.26** | **57.26** | **28.34** | **93.30** | **78.97** | **78.79 (+0.12)** | **53.19 (+6.41)** |
| Unseen | "a photo of my {label}" | TGA-ZSR | **72.68** | 43.94 | 80.61 | 39.61 | 50.97 | 20.18 | 90.52 | 73.12 | 74.03 | 44.30 |
| | | SAFT-L | 70.57 | **50.71** | **82.51** | **48.79** | **51.77** | **25.69** | **91.35** | **76.07** | **75.21 (+1.18)** | **50.18 (+5.88)** |
| | "a sketch of a {label}" | TGA-ZSR | **74.00** | 45.12 | 84.41 | 41.41 | 55.00 | 21.64 | 93.68 | 76.01 | 77.69 | 46.35 |
| | | SAFT-L | 71.97 | **51.73** | **84.71** | **51.34** | **55.41** | **27.58** | **94.12** | **80.05** | **78.08 (+0.39)** | **52.99 (+6.64)** |
| | "a drawing of a {label}" | TGA-ZSR | **74.42** | 45.55 | 84.86 | 42.17 | **55.89** | 22.24 | 93.44 | 76.05 | 78.06 | 46.82 |
| | | SAFT-L | 72.22 | **52.05** | **85.13** | **52.01** | 55.67 | **27.75** | **94.13** | **80.19** | **78.31 (+0.25)** | **53.32 (+6.50)** |
| | "a plastic {label}" | TGA-ZSR | **71.52** | 43.58 | 84.44 | 41.59 | **53.72** | 20.56 | 92.48 | 74.87 | **76.88** | 45.67 |
| | | SAFT-L | 69.50 | **49.93** | 83.12 | **50.00** | 53.22 | **26.41** | **92.69** | **78.32** | 76.35 (-0.54) | **51.58 (+5.91)** |
| | "a cartoon {label}" | TGA-ZSR | **72.10** | 43.64 | 85.55 | 43.23 | 53.61 | 21.57 | 92.84 | 75.52 | 77.33 | 46.77 |
| | | SAFT-L | 71.33 | **51.09** | **85.93** | **53.36** | **53.95** | **27.35** | **93.09** | **79.45** | **77.66 (+0.33)** | **53.39 (+6.62)** |
| | "a toy {label}" | TGA-ZSR | **70.56** | 42.51 | 84.58 | 44.45 | 51.93 | 20.50 | **90.54** | 72.63 | **75.68** | 45.86 |
| | | SAFT-L | 69.47 | **49.59** | 82.53 | **50.32** | **52.48** | **26.05** | 89.89 | **75.49** | 74.96 (-0.72) | **50.62 (+4.76)** |
| | "a photo of a cool {label}" | TGA-ZSR | **75.98** | 46.76 | 84.95 | 41.28 | 57.02 | 22.06 | 92.80 | 75.34 | 78.26 | 46.23 |
| | | SAFT-L | 73.51 | **53.02** | **85.83** | **51.57** | **57.10** | **27.62** | **93.49** | **79.56** | **78.81 (+0.55)** | **52.92 (+6.69)** |
| | "a photo of a clean {label}" | TGA-ZSR | **74.01** | 45.76 | 81.67 | 39.34 | 54.58 | 20.85 | 88.28 | 70.66 | 74.85 | 43.62 |
| | | SAFT-L | 71.87 | **51.70** | **83.03** | **49.20** | **55.43** | **26.92** | **89.57** | **73.86** | **76.01 (+1.16)** | **49.99 (+6.37)** |
| | "a tattoo of the {label}" | TGA-ZSR | **69.23** | 41.47 | 83.58 | 41.65 | 51.53 | 20.30 | 92.29 | 74.76 | 75.80 | 45.57 |
| | | SAFT-L | 67.44 | **48.27** | 83.23 | **50.91** | **52.74** | **26.91** | **93.63** | **79.76** | **76.53 (+0.73)** | **52.53 (+6.96)** |
| | "a photo of one {label}" | TGA-ZSR | **76.20** | 46.81 | 84.87 | 41.59 | 56.11 | 21.20 | 92.96 | 75.23 | 77.98 | 46.01 |
| | | SAFT-L | 73.49 | **52.85** | **85.16** | **51.87** | **56.50** | **27.38** | **93.54** | **79.08** | **78.40 (+0.42)** | **52.78 (+6.77)** |
| | "a black and white photo of the {label}" | TGA-ZSR | **72.11** | 43.82 | 84.81 | 40.71 | 54.42 | 20.92 | 92.45 | 74.38 | 77.22 | 45.34 |
| | | SAFT-L | 71.02 | **50.76** | **85.59** | **49.98** | **55.22** | **26.75** | **93.72** | **79.33** | **78.17 (+0.95)** | **52.02 (+6.68)** |
| | "a low resolution photo of the {label}" | TGA-ZSR | **73.85** | 44.24 | **85.14** | 40.44 | 55.62 | 21.45 | 92.44 | 75.01 | 77.73 | 45.63 |
| | | SAFT-L | 72.23 | **51.51** | 84.71 | **50.09** | **55.66** | **27.02** | **93.30** | **78.95** | **77.89 (+0.16)** | **52.02 (+6.39)** |

## A.5 Adaptability to Larger Perturbation Budget

Table 11: Robust accuracy (%) of different methods against PGD-100 ($\epsilon = 4/255$) across 16 datasets. Tiny-ImageNet is the source dataset and the others are zero-shot datasets. The best accuracy is highlighted in **bold** and the second-best accuracy is underlined. "Zero-shot Averag" refers to the averaged robust accuracy across 15 zero-shot datasets.

| Method | Source Tiny-ImageNet | CIFAR-10 | CIFAR-100 | Food101 | STL-10 | OxfordPets | Flowers102 | DTD | EuroSAT | FGVC-Aircraft | Caltech-101 | Caltech-256 | StanfordCars | PCAM | ImageNet-1K | SUN397 | Zero-shot Average |
|---|---|---|---|---|---|---|---|---|---|---|---|---|---|---|---|---|---|
| TeCoA | 1.63 | 0.29 | 0.45 | 0.25 | 9.29 | 1.25 | 0.50 | 2.23 | 0.00 | 0.00 | 11.18 | 5.80 | 0.04 | 2.53 | 0.58 | 0.26 | 2.31 |
| PMG-AFT | 4.94 | 3.77 | 3.50 | 1.38 | 19.54 | 2.73 | **2.85** | **3.99** | **0.47** | **0.06** | 20.33 | 10.00 | **0.29** | **24.33** | 2.38 | **1.51** | 6.47 |
| TGA-ZSR | **16.76** | 12.04 | 5.04 | **2.13** | 36.48 | 6.80 | 1.94 | 0.90 | 0.00 | 0.00 | 22.73 | 15.05 | 0.16 | 0.02 | **2.51** | 1.10 | 7.13 |
| SAFT-L | 15.92 | **19.09** | **6.87** | 1.70 | **43.40** | **7.20** | 1.14 | 0.59 | 0.00 | 0.00 | **27.63** | **19.15** | 0.10 | 0.01 | 2.11 | 0.78 | **8.65** |

## A.6 Generalizability to Out-of-domain Datasets

Table 12: Experiments on TU-Berlin sketch dataset and ImageNet-R. We report the best in **bold**.

| | Method | Sketch | ImageNet-R |
|---|---|---|---|
| Clean | TGA-ZSR | 36.80 | 46.74 |
| | SAFT-L | **43.85** | **52.85** |
| Robust | TGA-ZSR | 27.05 | 26.54 |
| | SAFT-L | **29.75** | **29.70** |

