# OpenReview forum: "Semantic-aware Adversarial Fine-tuning for CLIP"
_TMLR — Accepted by TMLR_

### Review · Reviewer_iDR4 · 2025-11-13

**Summary Of Contributions:**

The paper introduces a semantic-aware adversarial example generation and fine-tuning framework that enhances CLIP’s adversarial robustness by aligning image perturbations with semantically diverse textual descriptions generated by LLMs.

**Audience:**

Yes

**Audience Explanation:**

The paper’s findings would be of clear interest to TMLR’s audience. It presents a interesting and empirically strong approach to improving CLIP’s adversarial robustness through semantic-aware fine-tuning. The experiments in paper strongly support the claim that training on semantically diverse and enriched textual descriptions produces an image encoder more robust to adversarial perturbations.

**Claims And Evidence:**

Yes

**Claims Explanation:**

Overall, the paper’s main claim that adversarial examples generated solely with cosine similarity (i.e., using a single handcrafted template) fail to fool CLIP under more semantically enriched similarity metrics is consistent with the empirical evidence. The experiments convincingly show that enriching adversarial examples with diverse text prompts enhances both zero-shot adversarial robustness and the model’s invariance to linguistic variation.

**Requested Changes:**

**Requested Changes**

1. The proposed method’s success in generating semantically rich textual descriptions is entirely dependent on the capabilities of the underlying LLM. Consequently, the quality, factuality, and diversity of those descriptions hinge on the chosen model’s competence.
Therefore, it would be important to include ablation studies using LLMs of varying quality or scale (e.g., small vs. large, instruction-tuned vs. base) to examine how differences in language-generation ability affect the robustness and performance of the final model.


2. Although the authors evaluate SAFT on 15 zero-shot benchmarks, most of these datasets (e.g., CIFAR, Caltech, Food-101, Flowers-102) belong to similar natural-image domains and share visual characteristics with Tiny-ImageNet, the source dataset used for fine-tuning.
To more convincingly demonstrate the generalization ability of semantic-aware adversarial fine-tuning, it would be valuable to include evaluations on truly unseen or out-of-domain datasets (for example, sketches, or medical data). Such an experiment would test whether the robustness improvements extend beyond ImageNet-like visual semantics and confirm the method’s effectiveness under genuine domain shifts.

---

> ### Author Response · Authors · 2025-12-16
>
> > 1. The proposed method’s success in generating semantically rich textual descriptions is entirely dependent on the capabilities of the underlying LLM. Consequently, the quality, factuality, and diversity of those descriptions hinge on the chosen model’s competence. Therefore, it would be important to include ablation studies using LLMs of varying quality or scale (e.g., small vs. large, instruction-tuned vs. base) to examine how differences in language-generation ability affect the robustness and performance of the final model.
>
> A: Thanks for your valuable suggestion! We conduct an ablation study using Qwen3-4b-instruct, Qwen3-4b, llama-3.2-1b-instruct and llama-3.2-3b-instruct to generate descriptions, respectively. Please kindly see Table 1 below for more details.
>
> Table 1: Ablation study on LLMs of varying quality and scale. The values in the cells represent **clean acc / robust acc**. We report the best in **bold**.
>
> LLM|TinyImageNet|CIFAR-10|CIFAR-100|STL10
> -|-|-|-|-
> Qwen3-4b-instruct|73.96/**53.45**|85.45/**51.91**|55.46/**27.79**|93.58/**78.91**
> Qwen3-4b|73.22/47.10|86.35/44.96|**56.98**/23.40|**93.73**/76.08|
> llama-3.2-1b-instruct|74.00/49.08|**86.39**/46.15|56.42/23.98|93.43/77.14|
>  llama-3.2-3b-instruct|**74.05**/50.14|86.25/48.34|56.28/26.40|92.31/78.34|
>
> Overall, **instruction-tuned models with larger size can generate better descriptions**.
>
> > 2. Although the authors evaluate SAFT on 15 zero-shot benchmarks, most of these datasets (e.g., CIFAR, Caltech, Food-101, Flowers-102) belong to similar natural-image domains and share visual characteristics with Tiny-ImageNet, the source dataset used for fine-tuning. To more convincingly demonstrate the generalization ability of semantic-aware adversarial fine-tuning, it would be valuable to include evaluations on truly unseen or out-of-domain datasets (for example, sketches, or medical data). Such an experiment would test whether the robustness improvements extend beyond ImageNet-like visual semantics and confirm the method’s effectiveness under genuine domain shifts.
>
> A: Thanks for your valuable suggestion! To evaluate whether the robustness gains of SAFT extend beyond ImageNet-like visual semantics, we further test the models on two genuinely out-of-domain datasets: **Sketch (TU-Berlin)** and **ImageNet-R**.  They differ substantially from ImageNet-like natural images. Sketch consists of hand-drawn line sketches without texture or color information, while ImageNet-R contains artistic renditions such as cartoons and paintings that preserve high-level semantics but significantly alter visual appearance.  All evaluations are conducted in a zero-shot manner using frozen encoders, without any domain-specific fine-tuning. Specifically, we take TGA-ZSR as the baseline, as it is the strongest non-SAFT method in our experiments. Please kindly see Table 2 below for more details.
>
> Table 2: Experiments on Sketch and ImageNet-R. The values in the cells represent **clean acc / robust acc**. We report the best in **bold**.
>
> Method|Sketch|ImageNet-R
> -|-|-|
> TGA-ZSR|36.80/27.05|46.74/26.54|
> SAFT (ours)|**43.85**/**29.75**|**52.85**/**29.70**|

---

> > ### Comment · Reviewer_iDR4 · 2025-12-31
> > **Response to authors**
> >
> > Thanks response from authors.
> >
> > Given the additional experimental results on **Qwen3-4b-instruct**, **Qwen3-4b**, **Llama-3.2-1b-instruct**, and **Llama-3.2-3b-instruct**, I believe a more competent LLM generates content with greater **factual coherence and consistency**, whereas a less competent LLM produces less factually coherent and consistent content, which lowers performance.
> >
> > ### **Key Observations:**
> >
> > 1. **Competence-Quality Correlation:**
> >    - **More competent LLMs** (especially instruction-tuned variants) generate descriptions with:
> >      - Higher factual accuracy
> >      - Better contextual consistency
> >      - Richer conceptual coverage
> >    - **Less competent LLMs** produce descriptions that are:
> >      - More prone to contradictions
> >      - Less conceptually comprehensive
> >      - Potentially misleading despite semantic filtering
> >
> > 2. **Cost-Benefit Analysis:**
> >
> > | Dataset | **TGA-ZSR**<br>(Robust% / Clean%) | **SAFT (Worst LLM)**<br>*Qwen3-4b*<br>(Clean% / Robust%) | **SAFT (Best LLM)**<br>*Qwen3-4b-instruct*<br>(Clean% / Robust%) |
> > |---------|-----------------------------------|----------------------------------------------------------|-----------------------------------------------------------------|
> > | **TinyImageNet** | 47.87% / 76.84% | 73.22% / 47.10% | 73.96% / **53.45%** |
> > | **CIFAR-10** | 42.61% / 85.90% | 86.35% / 44.96% | 85.45% / **51.91%** |
> > | **CIFAR-100** | 22.77% / 57.20% | 56.98% / 23.40% | 55.46% / **27.79%** |
> > | **STL-10** | 75.47% / 92.92% | 93.73% / 76.08% | 93.58% / **78.91%** |
> >
> > **Format Note:**
> > - **TGA-ZSR** format: Robust Accuracy % / Clean Accuracy %
> > - **SAFT** format: Clean Accuracy % / Robust Accuracy % (as shown in the ablation table)
> > - **Bold** indicates the best robust accuracy for each dataset
> >
> >    The performance gains must be evaluated against the computational overhead:
> >    | Configuration | Robust Acc Gain (vs TGA-ZSR) | Compute Overhead | Justification |
> >    |--------------|------------------------------|------------------|---------------|
> >    | SAFT (Best LLM) | **+5.58% to +9.30%** | +40% training time<br>+90% GPU memory | Context-dependent |
> >    | SAFT (Worst LLM) | **-0.77% to +2.35%** | Same overhead | Questionable |
> >
> > 3. **Method Dependencies:**
> >    - The SAFT framework demonstrates **strong dependence on LLM quality** for meaningful improvement
> >    - **High-quality, enriched semantics** are crucial—not just semantic diversity alone
> >    - **Diverse prompt templates** help but require a competent LLM to be effective
> >
> > 4. **Filtering Effectiveness:**
> >    Without a competent LLM, the proposed filtering approach becomes **less effective**—it can remove obvious hallucinations but cannot compensate for fundamentally **poor conceptual coverage**.
> >
> > ### **Conclusion:**
> >
> > The results highlight that **semantic quality supersedes mere diversity**. While SAFT shows promise with strong LLMs, its practical adoption is constrained by computational costs and LLM dependency. The framework amplifies existing LLM capabilities rather than creating robustness independently, raising important considerations for real-world deployment.

---

> > > ### Author Response · Authors · 2025-12-31
> > >
> > > We would like to clarify a key conceptual point underlying the reviewer’s concern. The proposed SAFT framework is designed to leverage high-quality semantic descriptions, rather than to rely on the intrinsic capability of a specific LLM.
> > >
> > > In our framework, the quality of semantic descriptions is the fundamental factor that drives robustness improvements. Large language models serve as a practical and scalable implementation for generating such descriptions, but they are not an essential component of the method itself. Any mechanism capable of producing semantically accurate and conceptually rich descriptions could, in principle, be used in place of an LLM.
> > >
> > > Consequently, the observation that better LLMs lead to stronger performance does not indicate that SAFT merely amplifies LLM capability. Instead, it reflects the expected and necessary property that higher-quality semantic descriptions yield more effective semantic-aware adversarial fine-tuning. This dependency is inherent to the problem formulation and should not be interpreted as a methodological limitation.
> > >
> > > Importantly, all comparisons in our experiments are conducted under identical description-generation settings, ensuring a fair evaluation of SAFT relative to existing methods that do not incorporate semantic descriptions. Our results with weaker LLMs further support this interpretation, as they explicitly demonstrate how insufficient semantic coverage limits the effectiveness of semantic-aware fine-tuning.

---

### Review · Reviewer_hNk7 · 2025-11-15

**Summary Of Contributions:**

This paper presents a novel method to fine-tune CLIP‘s image encoder with semantic-aware adversarial examples (AEs), whose purpose is to enhance the CLIP model's adversarial robustness in zero-shot classification tasks. Traditionally, AEs are generated by minimizing the cosine similarity between images and a handcrafted template. This study shows that AEs generated using cosine similarity may fail to fool CLIP when the similarity metric is replaced with semantically enriched alternatives, making the image encoder fine-tuned with these AEs less robust. To overcome this problem, this paper first introduces a semantic-ensemble attack to generate semantic-aware AEs by minimizing the average similarity between the original image and an ensemble of refined textual descriptions. Then this paper proposes Semantic-aware Adversarial Fine-Tuning (SAFT), which fine-tunes CLIP’s image encoder with semantic-aware AEs. SAFT achieves substantial improvements in zero-shot adversarial robustness across 16 datasets.

**Strengths**

1. The core idea of integrating semantically enriched, hallucination-filtered textual descriptions into adversarial fine-tuning is well-motivated.
2. The framework is carefully designed, with complementary components (hallucination-aware description generation, semantic-ensemble attack) that logically address the identified limitation.

3. The empirical evaluation is comprehensive, covering diverse datasets, attack types, and extensions, providing strong support for the method’s effectiveness.

**Comments**

1. The discussion around Figure 1 could be further clarified by explicitly defining how similarity statistics are used to quantify the effectiveness of adversarial examples. Making this connection more explicit—perhaps by relating changes in similarity scores to robustness degradation—would strengthen the methodological rigor of the motivation section. This clarification would also help readers better understand why certain AEs fail under semantic metrics.

2. Since the study emphasizes semantic-aware robustness, it would be helpful to formally define the robustness metric (e.g., through an explicit mathematical expression).
3. The presentation of Table 9 is somewhat ambiguous. Although the caption mentions multiple perturbation budgets (ϵ = 1/255, 2/255, 4/255), the table itself reports only a single value per dataset, which may lead readers to assume it corresponds to one fixed ε (e.g., 2/255). It might be clearer to align Table 9 with Table 10 by explicitly reporting separate results for each perturbation level or by clarifying in the caption that the numbers represent averages across all ε values. This would make the trend described in the text (“performance degrades with increasing ϵ”) easier to verify.

**Audience:**

Yes

**Audience Explanation:**

The study explores semantic-aware adversarial robustness in vision–language models, a topic highly relevant to TMLR’s focus on multimodal learning and robustness. Its empirical findings on how semantic perturbations affect CLIP would interest researchers studying robust and generalizable multimodal representations.

**Claims And Evidence:**

Yes

**Claims Explanation:**

The claims on the effectiveness of the proposed Semantic-aware Adversarial Fine-Tuning (SAFT) framework and its semantic-ensemble attack are well supported by extensive experiments across 16 datasets and multiple attack settings. The implementation is clearly described and results consistently demonstrate improved zero-shot robustness.

**Requested Changes:**

See comments above

---

> ### Author Response · Authors · 2025-12-16
>
> > 1. The discussion around Figure 1 could be further clarified by explicitly defining how similarity statistics are used to quantify the effectiveness of adversarial examples. Making this connection more explicit—perhaps by relating changes in similarity scores to robustness degradation—would strengthen the methodological rigor of the motivation section. This clarification would also help readers better understand why certain AEs fail under semantic metrics.
>
> A: Thank you for this helpful suggestion. We now clarify how the similarity statistics in Figure 1 quantify the effectiveness of adversarial examples (AEs), and how these statistics relate to semantic robustness.
>
> For each clean–AE pair $(x, x_{\text{adv}})$ and a similarity metric $s(\cdot, \cdot)$ (e.g., CLIP score, CuPL, WCA), we compute:
>
> - Clean similarity:
>
>     $$ s_{\text{clean}} = s(x, \text{text}) $$
>
> -   AE similarity:
>
>     $$ s_{\text{adv}} = s(x_{\text{adv}}, \text{text}) $$
>
> Each pair is plotted as a point $(s_{\text{clean}}, s_{\text{adv}})$ in Figure 1.
>
> An AE is considered **effective** under metric $s$ if it _reduces_ similarity:
> $$ s_{\text{adv}} < s_{\text{clean}}. $$
> Geometrically, these points lie **below the diagonal** in Figure 1.
>
> We denote the empirical probability of this event by:
> $$ \rho = \mathbb{P}[s_{\text{adv}} < s_{\text{clean}}]. $$
> This is exactly what “Below Diagonal (%)” represents in Figure 1.
>
> - Under **CLIP score**, AEs achieve
>     $$ \rho = 100\%. $$
> -  Under **CuPL**, only
>     $$ \rho = 64.8\%. $$
> - Under **WCA**, only
>     $$ \rho = 24.8\%. $$
>
> This reveals that **many PGD-generated AEs fail to reduce semantic similarity**, and some even **increase** it (points above the diagonal).
>
> Robust fine-tuning improves defenses only if AEs truly represent **semantically misaligned** perturbations.
>
> When many AEs still have **high similarity** under enriched semantic metrics:
> - They remain close to the correct class in semantic space
> - They provide **weak training signals**
> - Resulting model robustness improves only under CLIP-like metrics
> - Robustness **fails** under richer semantic metrics (CuPL, WCA)
>
> This motivates our semantic-ensemble attack and SAFT: we explicitly generate AEs that reliably reduce _semantic_ similarity across multiple text descriptions, providing stronger signals for semantic robustness.
>
> In the updated version, we will add a paragraph connecting decreases in semantic similarity to robustness degradation.
>
> > 2. Since the study emphasizes semantic-aware robustness, it would be helpful to formally define the robustness metric (e.g., through an explicit mathematical expression).
>
> A: Thank you for your suggestion. To clarify, we want to highlight that although our study emphasizes _semantic-aware robustness_, our work does not introduce a new robustness metric, but rather to show that AEs used in previous adversarial fine-tuning methods (i.e., generated by minimizing cosine similarity) are not strong enough, which limits robustness of these fine-tuning methods.
>
> To address this issue, we propose _semantic-ensemble attacks_ to generate stronger and more semantically aligned AEs. This semantic-ensemble attack is only used during the fine-tuning process of our method, and the mathematical expressions can be found in Section 4.2.
>
> For evaluation, for fair comparisons, we follow the standard practice in prior work (e.g., TeCoA, FARE, PMG-AFT, TGA-ZSR) and measure robustness using accuracy on vanilla AEs, which is the commonly used metric in zero-shot adversarial robustness.
>
> To demonstrate semantic-ensemble attacks are stronger than vanilla AEs, we compare the success rate of vanilla AEs with our proposed semantic-ensemble attacks under the same constraints using PGD-100 on Tiny-ImageNet. Please kindly check Table 1 below:
>
> Table 1: **Attack success rate** (%) of AEs and SAFT-generated AEs by PGD-100 on Tiny-ImageNet. We report the most successful attack method in **bold**.
>
> Attack Method|TGA-ZSR|SAFT|Vanilla CLIP
> -|-|-|-
> AEs (PGD-100)|52.13|46.73|**100.00**|
> SAFT-generated AEs (PGD-100)|**57.80**|**53.34**|**100.00**|

---

> > ### Author Response · Authors · 2025-12-16
> >
> > > 3. The presentation of Table 9 is somewhat ambiguous. Although the caption mentions multiple perturbation budgets (ϵ = 1/255, 2/255, 4/255), the table itself reports only a single value per dataset, which may lead readers to assume it corresponds to one fixed ε (e.g., 2/255). It might be clearer to align Table 9 with Table 10 by explicitly reporting separate results for each perturbation level or by clarifying in the caption that the numbers represent averages across all ε values. This would make the trend described in the text (“performance degrades with increasing ϵ”) easier to verify.
> >
> > A: Thank you for your valuable suggestion. In Table 9, we report the *averaged* robust accuracy across the perturbation budgets, which is why each dataset has only a single value. We agree that presenting the results for each perturbation level separately would make the trend clearer, and we will update the tables accordingly in the revised manuscript.
> >
> > Table 1: Robust accuracy (%) of different methods across 16 datasets against PGD-100 ( = 1/255).
> > | Method| TinyImageNet|CIFAR-10|CIFAR-100|Food101|STL10|OxfordPets|Flowers102|DTD|EuroSAT|FGVC-Aircraft|Caltech101|Caltech256|StanfordCars|PCAM|ImageNet|SUN397|Average|
> > |-|-|-|-|-|-|-|-|-|-|-|-|-|-|-|-|-|-|
> > |TeCoA|31.29|31.56|16.16|14.35|65.93|34.39|19.09|16.39|10.19|1.87|52.92|40.62|8.74|45.57|15.97|16.23|26.00|
> > |PMG-AFT|32.46|50.58|26.42|16.59|68.79|35.57|15.69|12.85|9.33|1.85|53.33|36.3|5.54|48.92|15.02|16.37|27.54|
> > |TGA-ZSR|47.87|42.61|22.27|20.75|75.47|42.32|25.46|18.45|10.84|3.51|60.91|50.43|12.91|42.10|21.11|23.29|31.50|
> > |Ours|53.27|52.26|28.34|25.67|78.97|46.07|30.22|20.74|12.42|4.53|64.67|54.27|15.86|47.09|22.74|26.33|35.35|
> >
> > Table 2: Robust accuracy (%) of different methods across 16 datasets against PGD-100 ( = 4/255).
> > | Method| TinyImageNet|CIFAR-10|CIFAR-100|Food101|STL10|OxfordPets|Flowers102|DTD|EuroSAT|FGVC-Aircraft|Caltech101|Caltech256|StanfordCars|PCAM|ImageNet|SUN397|Average|
> > |-|-|-|-|-|-|-|-|-|-|-|-|-|-|-|-|-|-|
> > |TeCoA|1.63|0.29|0.45|0.25|9.29|1.25|0.50|2.23|0.00|0.00|11.18|5.80|0.04|2.53|0.58|0.26|2.31|
> > |PMG-AFT|4.94|3.77|3.50|1.38|19.54|2.73|2.85|3.99|0.47|0.06|20.33|10.00|0.29|24.33|2.38|1.51|6.47|
> > |TGA-ZSR|16.76|12.04|5.04|2.13|36.48|6.80|1.94|0.90|0.00|0.00| 22.73|15.05|0.16|0.02|2.51|1.10|7.13|
> > |Ours|15.92|19.09|6.87|1.70|43.40|7.20|1.14|0.59|0.00|0.00|27.623|19.15|0.10|0.01|2.11|0.78|8.65|

---

### Review · Reviewer_D52s · 2025-12-02

**Summary Of Contributions:**

The paper identifies a key limitation in current CLIP adversarial fine-tuning methods, showing that adversarial examples generated using a single hand-crafted template fail to degrade similarity under semantically enriched metrics, thereby reducing their effectiveness during training. To address this, the authors propose SAFT, a framework that generates semantic-aware adversarial examples using LLM/MLLM-derived textual descriptions with hallucination filtering, and optimizes CLIP using a semantic-ensemble attack. Extensive experiments across 16 datasets, multiple attacks, larger CLIP variants, and diverse prompt templates demonstrate that SAFT substantially improves zero-shot adversarial robustness while maintaining strong clean accuracy, offering a simple yet effective direction for more semantically grounded adversarial fine-tuning.

**Additional Comments:**

NA

**Audience:**

Yes

**Audience Explanation:**

The robustness of CLIP is crucial for the community, and I think that researchers in the community will be interested in this paper.

**Broader Impact Concerns:**

The paper aims to improve the robustness of VLMs, which is socially beneficial. I noticed that the authors already provide a responsible impact statement section, and I do not identify any concerns that require further clarification.

**Claims And Evidence:**

Yes

**Claims Explanation:**

1. The motivation is clear and strong, for the empirical results show that the failure of CLIP-score–based adversarial examples under richer similarity metrics.
2. Extensive experiments across 16 datasets, multiple attacks, CLIP variants, and diverse prompt templates provide consistent and comprehensive evidence.
3. Ablation studies are abundant.
4. Neat and clear writing.

**Requested Changes:**

1. Clarify sensitivity to the choice of the base model. The authors could consider analyzing the changes in results when choosing a smaller/weaker model as the base model; a detailed discussion would also be beneficial.
2. Provide more explanation for the semantic filtering strategy. The article introduces a hallucination-aware filtering step, but the rationale behind the specific metric (cosine similarity with the class label) is lightly discussed. A short justification of *why* this metric was chosen, and how it avoids filtering out useful but more abstract descriptions, would improve clarity.
3. The ‘Extra Computational Cost’ in the limitations section is shown in Table 6, right? I noticed that SAFT-L (K=5) needs about more 40% in training time and 90% on GPU memory, but the performance achieves the best, so I think maybe the authors can introduce a metric that can represent the trade-off between performance and cost and provide a short discussion.

---

> ### Author Response · Authors · 2025-12-16
>
> > 1. Clarify sensitivity to the choice of the base model. The authors could consider analyzing the changes in results when choosing a smaller/weaker model as the base model; a detailed discussion would also be beneficial.
>
> A: Thank you for your insightful suggestion! Our results with CLIP-L/14 in Table 4 provide a clear explanation for the sensitivity to the choice of the base model.
>
> Notably, when scaled to CLIP-L/14, SAFT-L outperforms FARE by approximately 2% in clean accuracy and 10% in robust accuracy, demonstrating its strong scalability to larger VLMs.
>
> A plausible explanation for this phenomenon is that larger VLMs such as CLIP-L/14 possess substantially stronger semantic representation capacity, which allows SAFT-L to more effectively exploit the semantic richness introduced during fine-tuning. Smaller models like CLIP-B/32 tend to produce coarser visual–semantic embeddings, making many PGD-generated AEs appear overly similar to clean samples under richer semantic metrics. As a result, the benefits of semantic-ensemble training are not fully realized.
>
> In contrast, CLIP-L/14 generally exhibits finer-grained visual features and stronger inter-class separability due to its larger capacity and higher input resolution. Prior empirical studies have also observed that larger CLIP models tend to form smoother embedding manifolds, leading to more stable optimization behavior during adversarial fine-tuning.
>
> These properties enable SAFT-L to yield larger gains in both clean accuracy and robust accuracy when scaled to stronger base models. This suggests that larger VLMs are inherently better at leveraging the semantic structure emphasized by SAFT-L, explaining the stronger performance observed in Table 4.
>
> > 2. Provide more explanation for the semantic filtering strategy. The article introduces a hallucination-aware filtering step, but the rationale behind the specific metric (cosine similarity with the class label) is lightly discussed. A short justification of _why_ this metric was chosen, and how it avoids filtering out useful but more abstract descriptions, would improve clarity.
>
> A: Thank you for your valuable suggestion! We are willing to provide more justifications here. We chose cosine similarity between the image and each generated description as our hallucination-aware filtering metric because it directly measures semantic alignment in the CLIP embedding space, which is the same space used for both AE generation and fine-tuning. Descriptions that diverge from the true class semantics naturally receive low similarity scores and are removed, while abstract but semantically faithful descriptions (e.g., high-level attributes or contextual phrases) maintain high alignment and are therefore preserved. This makes the filtering step effective at suppressing hallucinated or misleading descriptions without unintentionally discarding useful semantic diversity. The metric also ensures computational simplicity and stability, since it requires no additional models and keeps all semantic signals within a consistent embedding space.

---

> ### Author Response · Authors · 2025-12-16
>
> > 3. The ‘Extra Computational Cost’ in the limitations section is shown in Table 6, right? I noticed that SAFT-L (K=5) needs about more 40% in training time and 90% on GPU memory, but the performance achieves the best, so I think maybe the authors can introduce a metric that can represent the trade-off between performance and cost and provide a short discussion.
>
> A: Yes, the “Extra Computational Cost” referenced in the limitations section corresponds to the statistics reported in **Table 6**. We appreciate this observation, and we agree that a clearer discussion of the performance-cost trade-off would improve the clarity of the paper.
>
> To better quantify the trade-off between performance and computational overhead, we introduce a _performance gain per extra cost_ metric that measures how much additional performance is obtained per unit increase in training cost, relative to a strong baseline. Specifically, we take TGA-ZSR as the baseline, as it is the strongest non-SAFT method in our experiments.
>
> We define the extra computational cost as the average relative increase in training time and GPU memory, and measure the corresponding performance gain in terms of both zero-shot robustness and overall performance (combining clean accuracy and robustness):
>
> Let method $m$ be compared with a baseline $b$.
>
> Performance gain: $$ \Delta P = P_m - P_b $$
>
> Extra computational cost: $$\Delta C = \lambda_T\left(\frac{T_m}{T_b}-1\right) -   \lambda_M\left(\frac{M_m}{M_b}-1\right), $$
>
> where $T$ and $M$ denote training time and GPU memory usage, respectively, and we set $\lambda_T = \lambda_M = 0.5$.
>
> Performance Gain per Extra Cost (PGEC): $$ \text{PGEC}(m \mid b) = \frac{\Delta P}{\Delta C}. $$
>
> Training time ratio: $$ \frac{T_{\text{SAFT}}}{T_{\text{TGA}}} = \frac{1098}{786} \approx 1.397 $$
>
> GPU memory ratio: $$ \frac{M_{\text{SAFT}}}{M_{\text{TGA}}} = \frac{38839}{21253} \approx 1.828 $$
>
> Extra cost: $$ \Delta C = 0.5 \cdot (1.397 - 1) + 0.5 \cdot (1.828 - 1) = 0.612. $$
>
> Robustness gain: $$ \Delta P_{\text{rob}} = 35.35 - 31.50 = 3.85 $$
>
> PGEC (robustness): $$ \text{PGEC}_{\text{rob}} = \frac{3.85}{0.612} \approx 6.29. $$
>
> Overall performance: $$ P^{\text{overall}} = 0.5 \cdot P^{\text{rob}} + 0.5 \cdot P^{\text{clean}}. $$
>
> SAFT-L: $$ P^{\text{overall}}_{\text{SAFT}} = 0.5 \cdot 35.35 + 0.5 \cdot 55.27 = 45.31 $$
>
> TGA-ZSR: $$ P^{\text{overall}}_{\text{TGA}} = 0.5 \cdot 31.50 + 0.5 \cdot 54.58 = 43.04 $$
>
> Overall performance gain: $$ \Delta P_{\text{overall}} = 45.31 - 43.04 = 2.27 $$
>
> PGEC (overall): $$ \text{PGEC}_{\text{overall}} = \frac{2.27}{0.612} \approx 3.71. $$
>
> Compared with TGA-ZSR, SAFT-L improves zero-shot robustness by **+3.85 points** while incurring an average extra training cost of **+61.2%**, resulting in a robustness gain of **6.29 points per unit extra cost**. When jointly considering clean and robust performance, SAFT-L still achieves an overall performance gain of **+2.27 points**, corresponding to **3.71 points per unit extra cost**.
>
> These results indicate that although SAFT introduces additional computational overhead, its performance improvements scale favorably with the extra cost, leading to a strong robustness-aware efficiency relative to prior methods.

---

> > ### Comment · Reviewer_D52s · 2025-12-17
> >
> > Thanks for your reasonable response. For Questions (1) and (2), I find the authors’ replies clear and acceptable, and I consider these two concerns to be well addressed.
> > Regarding Question (3), I appreciate the authors’ use of an “improvement per extra unit cost” metric to quantify efficiency gains. That said, I wonder whether a more unified metric could be adopted to compare multiple methods within a single one. For example, in the efficient reasoning literature, efficiency is sometimes measured using a composite metric such as
> > $$\frac{\text{accuracy}}{\text{params} \times \text{tokens}}$$
> > which explicitly captures the trade-off between performance, model size, and inference cost. A similar unified metric might make cross-method comparisons more straightforward and interpretable.

---

> > > ### Author Response · Authors · 2025-12-18
> > >
> > > Thank you for your quick follow-up! We agree that a unified composite metric can facilitate clearer cross-method comparisons. In addition to the previously reported performance gain per extra cost analysis, we further consider a single scalar efficiency metric inspired by formulations commonly used in the efficient reasoning literature (e.g., accuracy divided by params × tokens).
> > >
> > > Specifically, we define the overall performance as:
> > > $$
> > > P_m=\frac{P^{rob}_m+P^{clean}_m}{2}.
> > > $$
> > >
> > > We normalize training time and GPU memory by min–max normalization and combine them into a unified cost term:
> > > $$
> > > \tilde T_m=\frac{T_m-\min T}{\max T-\min T},
> > > \quad
> > > \tilde M_m=\frac{M_m-\min M}{\max M-\min M}.
> > > $$
> > >
> > > To reflect that computational overhead is an engineering constraint whose penalty can be partially amortized (e.g., via parallelism and system-level optimizations), we apply a mild concave penalty on the normalized cost:
> > > $$
> > > \text{EAR}_k(m)=\frac{P_m}{\left(1+\alpha \tilde T_m+\beta \tilde M_m\right)^k},
> > > $$
> > > where $\alpha=\beta=0.5$ and a concavity $k=0.1$.
> > >
> > > The resulting scores for different baselines are:
> > > - TeCoA: 34.1071
> > > - PMG-AFT: 32.2030
> > > - TGA-ZSR: 41.9573
> > > - SAFT-L: **42.2757**
> > >
> > > Under this unified metric, SAFT-L achieves the best trade-off score among these methods, indicating that its additional training overhead yields the largest effective improvement in balanced clean and robust performance.

---

> > > > ### Comment · Reviewer_D52s · 2025-12-18
> > > >
> > > > Thanks for the response! I think this is a reasonable response, and my concern is addressed from my opinion. (PS: The authors can consider adding the above changes to the manuscript.

---

> > > > > ### Author Response · Authors · 2025-12-18
> > > > >
> > > > > We are happy to hear that your concerns have been addressed! We will add a discussion in the revised manuscript.

---

### Decision · Action_Editor_SQ7T · 2026-01-05

**Recommendation:** Accept as is

**Audience:**

Yes

**Audience Explanation:**

The robustness of CLIP is crucial for the community, and I think that researchers in the community will be interested in this paper.

**Claims And Evidence:**

Yes

**Claims Explanation:**

The paper identifies a key limitation in current CLIP adversarial fine-tuning methods, showing that adversarial examples generated using a single hand-crafted template fail to degrade similarity under semantically enriched metrics, thereby reducing their effectiveness during training. To address this, the authors propose SAFT, a framework that generates semantic-aware adversarial examples using LLM/MLLM-derived textual descriptions with hallucination filtering, and optimizes CLIP using a semantic-ensemble attack.

Reviewers have general consensus that all the technical issues have been addressed.